# SenseShift6D: Multimodal RGB-D Benchmarking for Robust 6D Pose Estimation across Environment and Sensor Variations

## Abstract

Recent advances on 6D object-pose estimation have achieved high performance on representative benchmarks such as LM-O, YCB-V, and T-Less. However, these datasets were captured under fixed illumination and camera settings, leaving the impact of real-world variations in illumination, exposure, gain or depth-sensor mode—and the potential of test-time sensor control to mitigate such variations—largely unexplored. To bridge this gap, we introduce *SenseShift6D*, the first RGB-D dataset that physically sweeps 13 RGB exposures, 9 RGB gains, auto-exposure, 4 depth-capture modes, and 5 illumination levels. For six common household objects (spray, pringles, tincase, sandwich, mouse, and duck), we acquire 198.8k RGB and 20.0k depth images, which can provide 1,380 unique sensor-lighting permutations per object pose. Experiments with state-of-the-art models on our dataset demonstrate that applying multimodal sensor control at test time yields substantial performance gains, achieving a 18.0 pp improvement on pretrained generalizable models. It also enhances robustness precisely where those models tend to fail. Moreover, even instance-level pose estimators, where train and test set share identical object and background, performance still varies under environmental and sensor change, demonstrating that test-time sensor control remains effective compared to costly expansions in the quantity and diversity of real-world training data, without any additional training. *SenseShift6D* extends the object pose evaluation paradigm from data-centered to sensor-aware robustness, laying a foundation for adaptive, self-tuning perception systems capable of operating robustly in uncertain real-world environments.

## 1 Introduction

Estimating the 6D pose of everyday objects is a critical computer vision task, enabling fine-grained interactions in mixed-reality headsets, autonomous-vehicle manipulation, and embodied AI agents navigating cluttered environments (Doughty & Ghugre, 2022; Zhang et al., 2023; Trojak et al., 2025; Wu et al., 2019; Ke et al., 2020; Ponimatkin et al., 2025). The field has made substantial progress, with state-of-the-art methods achieving high accuracy on established benchmarks (Hinterstoisser et al., 2012a; Brachmann et al., 2014; Hodan et al., 2017; Kaskman et al., 2019; Hodan et al., 2018b).

However, these impressive results are achieved under **strictly controlled acquisition settings**. Real-world deployments, in contrast, face substantial variations: ambient illumination changes throughout the day; camera exposure and gain are adjusted by on-board auto-control or user preference; and commodity depth sensors switch capture modes on purpose. These environmental and sensor dynamics introduce complex artifacts—nonlinear color shifts and structured depth noise—that are difficult to reproduce via synthetic augmentations or compensate for during post-processing (Baek et al., 2024). Consequently, existing benchmarks—mostly collected with fixed illumination, exposure, gain, and depth configurations—cannot adequately represent real-world scenarios. Therefore, the research community lacks a systematic benchmark to quantify how 6D-pose estimators respond to per-frame sensor variation or to evaluate emerging test-time sensor-control strategies (Baek et al., 2024; 2025) designed to mitigate these shifts.

Recent attempts to tackle this robustness gap have explored adaptive sensor control, which dynamically adjusts sensor parameters at test time to maximize accuracy without modifying the underlying model. Studies in image classification have demonstrated that test-time sensor control improves

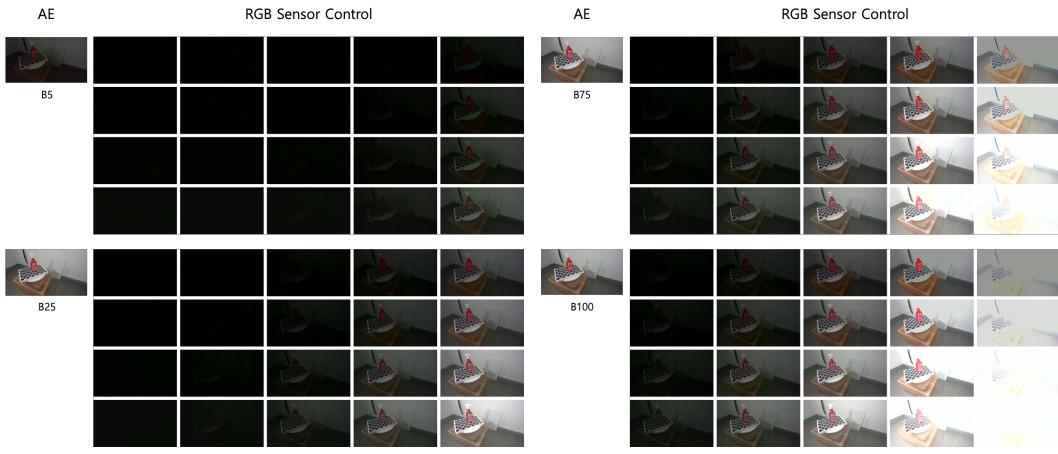

Figure 1: **RGB sample images under auto exposure and under all combinations of exposure and gain settings for brightness 5%, 25%, 75% and 100%.** Rows indicate gain levels and columns indicate exposure levels.

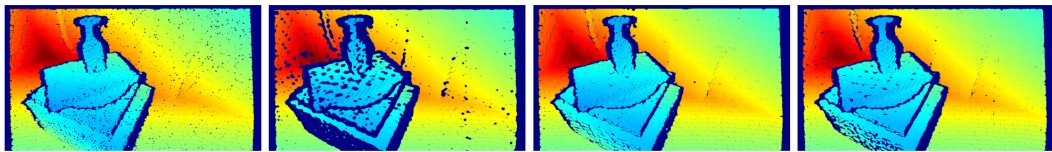

Figure 2: **Depth sample images under four depth capture modes:** default, high accuracy, high density, and medium density.

model robustness more effectively than scaling up the model size or augmenting training data (Baek et al., 2024; 2025). However, these efforts have been limited to unimodal RGB tasks. Importantly, no prior work has examined multimodal (RGB and depth) sensor control or provided a benchmark that physically sweeps multimodal sensor parameters with ground-truth 6D pose. Consequently, the effectiveness of adaptive multimodal sensing—and its interplay with traditional data-centric training—remains an open question.

To bridge this gap, we introduce *SenseShift6D*, a novel RGB-D benchmark for 6D pose estimation that **physically sweeps key sensor and lighting parameters**. Using an Intel RealSense RGB-D camera in a controlled darkroom setup, we systematically vary 13 exposure levels, 9 gain settings, 4 depth-capture modes, and 5 illumination conditions across six representative household objects: a spray bottle, a Pringles tube, a tincase, a sandwich replica, a vertical mouse, and a rubber duck. This protocol produces a dataset of 198.8k RGB and 20.0k depth frames, which can provide 1,380 unique sensor-lighting permutations per object pose (Figures 1 and 2), and 795.2k paired RGB-D scenes—to our knowledge, **the largest** of its kind. Each scene is annotated via 6D ground-truth via ArUco calibration and manual refinement (with ∼2 mm errors).

For controlled study, experiments in this paper are conducted on the four subsets of the entire dataset: **Train-Def**, captured under default camera and illumination settings, and **Train-Var**, covering a comprehensive multimodal parameter grid (8 exposure, 5 gain, auto-exposure, 1 depth-capture settings with 5 lighting conditions). Correspondingly, we construct two test splits (**Test-Def** and **Test-Var**) without overlap with the training configurations. As all variations are physically realized, the captured images preserve authentic sensor artifacts, including nonlinear color shifts, photon-shot noise, and structured depth errors, that synthetic augmentations cannot replicate. Thus, *SenseShift6D* establishes the first realistic testbed for sensor-aware 6D pose estimation and adaptive sensing.

This paper offers three main contributions:

- **Sensor-aware multimodal benchmarking:** We present the first 6D object-pose benchmark that systematically and physically varies multimodal sensor parameters across frames, enabling fine-grained analysis of sensor-dependent performance degradation.
- **New Axes of Generalization:** We reveal that environmental illumination and sensor settings constitute an orthogonal axis of generalization that current approaches do not address: (i) pretrained, generalizable models targeting unseen objects, occlusion, and clutter still degrade markedly under

Table 1: **Comparison of 6D pose estimation datasets in terms of sensor and lighting variation.**

| Dataset | Environmental and sensor variations | | | | | Objects | RGB | Depth | RGB-D |
|---|---|---|---|---|---|---|---|---|---|
| | Illumination | Exposure | Gain | Depth | Total Var. | | | | |
| LM (Hinterstoisser et al., 2012a) | × | × | × | × | × | 15 | 18.2k | 18.2k | 18.2k |
| LM-O (Brachmann et al., 2014) | × | × | × | × | × | 8 | 1.2k | 1.2k | 1.2k |
| T-LESS (Hodan et al., 2017) | × | × | × | × | × | 30 | 49k | 49k | 49k |
| YCB-V (Xiang et al., 2017) | × | × | × | × | × | 21 | 133k | **133k** | 133k |
| TUD-L (Hodan et al., 2018b) | **8 options** | × | × | × | 8 | 3 | 62k | 62k | 62k |
| HB (Kaskman et al., 2019) | 2 options | × | × | × | 2 | **33** | 34.8k | 34.8k | 34.8k |
| HOPE (Sundermeyer et al., 2022) | 5 options | × | × | × | 5 | 28 | 238 | 238 | 238 |
| IPD (Kalra et al., 2024) | 3 options | 4 levels | × | × | 12 | 20 | 30k | 30k | 30k |
| **SenseShift6D (Ours)** | 5 levels | **13 levels** | **9 levels** | **4 modes** | **1,380** | 6 | **198.8k** | **20.0k** | **795.2k** |

these shifts; and (2) instance-level models trained with standard augmentations likewise degrade even for known objects.

- **Robustness via multimodal test-time sensor control:** We demonstrate that test-time sensor control substantially improves performance *without model retraining*. While single-modal control is *already* effective (e.g., **+11.4 percentage points (pp)** on pretrained models), multimodal control delivers further gains (e.g., **+18.0 pp** on the same models), highlighting the *synergy* from jointly adapting RGB and depth sensors.

By providing a structured testbed for sensor-aware object 6D pose estimation, *SenseShift6D* lays the groundwork for more robust, adaptive perception systems in dynamic real-world environments.

## 2 RELATED WORK

### 2.1 6D POSE ESTIMATION BENCHMARK

**Classic datasets.** LineMOD (LM) (Hinterstoisser et al., 2012a) and its occlusion subset (LM-O) (Brachmann et al., 2014) pioneered RGB-D pose estimation benchmarks in controlled tabletop setups. Subsequent benchmarks such as T-LESS (Hodan et al., 2017), which provided a large collection of texture-less industrial objects, and YCB-V (Xiang et al., 2017), which offered cluttered scenes with temporal sequences, expanded the scope of evaluation. While these benchmarks provide robust evaluation regarding variations in objects and their poses, all were captured under fixed illumination and camera settings.

**Initial exploration of environmental and sensor variation.** TUD-L (Hodan et al., 2018b), Home-brewedDB (HB) (Kaskman et al., 2019), and HOPE (Sundermeyer et al., 2022) introduced variations in illumination conditions but retained fixed camera settings with auto-exposure enabled. The recent Industrial Parts Dataset (IPD) (Kalra et al., 2024) was the first benchmark to vary camera parameters, capturing four exposure levels under multiple lighting conditions. However, IPD focuses on texture-less industrial parts and does not explore variations in RGB gain or depth-sensor modes, thus limiting its suitability for adaptive-sensing research.

**Remaining gap and our contribution.** As summarized in Table 1, no existing benchmark offers orthogonal sweeps of RGB exposure, RGB gain, and multiple depth-capture modes together with dense 6D ground truth. Our *SenseShift6D* dataset directly addresses this gap by systematically varying exposure, gain, and depth-capture modes under diverse illumination conditions, enabling fine-grained analysis of 6D pose estimation robustness to realistic hardware-level changes.

### 2.2 ADAPTIVE SENSOR CONTROL BENCHMARK

Prior studies (Goudreault et al., 2023; Paul et al., 2022) have explored test-time adaptation of camera parameters to improve downstream vision performance, but these studies lacked explicit benchmarks in real-world. Recently, ImageNet-ES (Baek et al., 2024) and ImageNet-ES-Diverse (Baek et al., 2025) captured real-world scenes using a *physical* camera, systematically varying aperture, shutter speed, and ISO across multiple lighting conditions for image classification tasks. These benchmarks demonstrated that physical variation *cannot be replicated by synthetic augmentations* and dynamically selecting optimal camera parameters at test time can significantly increase the performance of lightweight models, enabling accuracy competitive with larger foundation models without additional training.

Table 2: **SenseShift6D Configurations for Data split.**

| Data split | Brightness (%) | Sensor configuration | Exposure ($\mu$s) | Gain | Depth-capture mode | Captured images |
|---|---|---|---|---|---|---|
| Train-Def | 50 | Auto (1 RGB options, 1 depth option) | Auto | Auto | Default | RGB: 750, Depth: 750 1 RGB-D scene/pose |
| Train-Var | 5, 25, 50, 75, 100 | Auto + Manual (41 RGB options, 1 depth options) | Auto, 2, 4, 19, 78, 312, 1250, 5000, 10000 | Auto, 0, 32, 64, 96, 128 | Default | RGB: 153.7k, Depth: 3.7k 225 RGB-D scenes/pose |
| Test-Def | 50 | Auto (1 RGB option, 1 depth option) | Auto | Auto | Default | RGB: 251, Depth: 251 1 RGB-D scenes/pose |
| Test-Var | 5, 25, 50, 75, 100 | Auto + Manual (21 RGB options, 4 Depth options) | Auto, 9, 39, 156, 625, 2500 | Auto, 16, 48, 80, 112 | Default, High Accuracy, High Density, Medium Density | RGB: 26.3k, Depth: 5.0k 480 RGB-D scenes/pose |

However, existing adaptive sensing research remains limited to single-modal RGB classification. To date, no prior work has explored: (1) multimodal sensor control, where depth-sensor modes must be jointly optimized with RGB settings; or (2) the impact of adaptive sensor control on object-centric 6D pose estimation tasks, which inherently rely on both photometric and geometric cues. Our *SenseShift6D* addresses this gap by providing the first realistic testbed for evaluating multimodal, per-frame sensor-control policies alongside conventional data-centric training baselines.

## 3 SENSESHIFT6D

We introduce *SenseShift6D*, the first RGB-D benchmark that *physically* sweeps camera and lighting parameters, enabling rigorous evaluation of 6D object pose estimators under realistic environmental and sensor dynamics. Using an Intel RealSense D455 mounted in a darkroom, we systematically capture six representative household objects—spray, Pringles, tincase, sandwich, mouse, and duck—selected to reflect diverse combinations of diverse RGB and geometric features (Figure 3), under auto-exposure, 13 RGB-exposure times, 9 RGB-gain settings, 4 depth-capture presets, and 5 illumination levels (details in Table 2). Our dataset consists of 198.8k RGB and 20.0k depth frames, which can yield 1,380 unique sensor-lighting permuta-

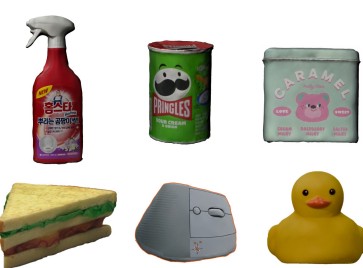

Figure 3: **Six household objects collected in SenseShift6D.**

tions per object pose and 795.2k RGB-D combined scenes, the *largest* among the current benchamrks to our knowledge. Accurate ground truth is obtained via a rigidly mounted ChArUco board and careful hand-eye calibration.

To facilitate controlled benchmarking, we organize the dataset into four clearly defined splits:

- **Train-Def:** Contains only the default sensor and lighting configuration (1 RGB-D scene/pose)
- **Train-Var:** Expands Train-Def, covering (8 exposures $\times$ 5 gains + 5 auto-exposure) $\times$ 1 depth modes $\times$ 5 brightness levels (225 RGB-D scenes/pose).
- **Test-Def:** Includes the default sensor and lighting configuration (1 RGB-D scenes/pose)
- **Test-Var:** Expands Test-Def, featuring (5 exposures $\times$ 4 gains + 4 auto-exposure) configurations, spanning the same depth-mode and illumination range (480 RGB-D scenes/pose).

As all variations are physically captured rather than digitally simulated, the resulting images inherently retain authentic artifacts, such as non-linear color shifts, photon-shot noise, and depth-mode-specific errors, that synthetic augmentation methods cannot reproduce (Baek et al., 2024). *SenseShift6D* therefore offers the first realistic testbed for analyzing environment- and sensor-aware robustness, enabling side-by-side evaluation of per-frame adaptive-sensing policies against traditional data-centric training strategies.

### 3.1 ENVIRONMENTAL AND SENSOR VARIATIONS

**Ambient-illumination sweep.** Real deployments frequently encounter lighting changes, such as those found in warehouse aisles, outdoor loading bays, or dim domestic environments. To reproduce these diverse conditions in a controlled and repeatable manner, we built a darkroom enclosed with black curtains to block external light completely. Within this setup, we installed three Philips Hue White & Color Ambiance bulbs, providing precise and programmable brightness control. Each bulb was calibrated to a consistent color temperature of 6535 K, with brightness systematically

varied across five levels: 5%, 25%, 50%, 75% and 100%. This controlled setting ensures uniform illumination conditions, enabling accurate comparisons of RGB gain and exposure settings under identical photon flux.

**RGB-sensor parameters.** The Intel RealSense D455 camera permits manual control of two key RGB sensor parameters: exposure, determining the duration of sensor exposure to incoming light, and gain, which amplifies the captured signal strength.

- **Exposure grid.** The D455 supports exposure settings ranging from 1 $\mu$s to 10,000 $\mu$s. Empirical tests indicated that exposure times below 1,000 $\mu$s yield image quality comparable to the camera's auto-exposure setting in well-lit environments. Given that this threshold is relatively low within the configurable range, we chose 13 logarithmically spaced exposure values to maximize coverage across practical operating conditions.
- **Gain grid.** The RGB gain of the D455 can be set from 0 to 128. Through visual inspection, we found that linear steps produce more diverse characteristics than exponential spacing. Therefore, we selected 9 linearly spaced gain settings spanning the full range.

**Depth capture modes.** The Intel RealSense D455 generates depth maps via stereo matching, estimating depth by aligning left and right camera images. This process relies on multiple internal parameters whose individual effects are complex and challenging to fine-tune directly. To manage this complexity, we leverage four predefined depth presets provided by Intel, each optimized for different usage scenarios (Hsu et al., 2023):

- **Default:** Offers visually clean depth maps with reduced noise and well-defined edges, suitable for general-purpose applications.
- **High Accuracy:** Provides highly reliable depth estimates by enforcing stricter confidence thresholds, albeit at the expense of a lower fill rate.
- **High Density:** Increases the fill factor, providing more complete depth maps, particularly effective for detecting more surfaces in low-texture areas.
- **Medium Density:** Offers a balanced trade-off between fill factor and accuracy, aiming to balance depth map completeness and precision.

### 3.2 Dataset Collection

To capture each scene systematically, we acquired images under all sensor configurations associated with a given data split, while maintaining fixed positions for the object and camera. Scenes were assigned exclusively to specific splits, ensuring no overlap between training and test sets. To capture diverse object poses, we used a motorized turntable to vary object orientation and adjusted the camera's position and tilt angle. Each object was rigidly mounted onto a ChArUco board, which itself was firmly placed on the turntable, which provides a consistent and precise pose reference throughout data capture.

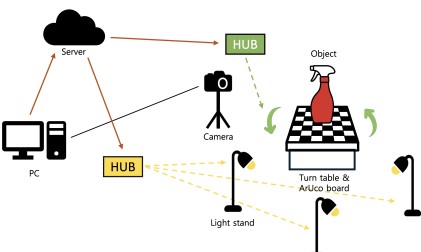

Figure 4: **Testbed for data collection.**

Table 3: **GT annotation accuracy (mm).**

|  | $\mu_\delta$ | $\sigma_\delta$ | $\mu_{|\delta|}$ | $med_{|\delta|}$ |
|---|---|---|---|---|
| RealSense | -0.28 | 3.29 | 2.32 | 1.73 |

For automatic collection, the turntable was remotely controlled, and deliberate delays were introduced after changing lighting brightness or sensor parameters, allowing sufficient stabilization time. We employed one smart IR remote-control hub to send commands to the turntable and another hub to control three smart bulbs. Both hubs received commands from a cloud-based server, which coordinated the entire data-capture sequence through an integrated control script (see Figure 4).

### 3.3 Data Processing

**Validation.** We manually reviewed the images for each parameter configuration. If any scene appeared inconsistent due to lighting or sensor parameter change delays, we excluded that scene across all configurations to maintain dataset consistency.

**Stabilizing auto-exposure frames.** The default setting to capture RGB images is the auto-exposure mode. For the D455 camera, we observed that the auto-exposure setting depends on the gain value set

in the previous frame, leading to frame-to-frame brightness jitter. To suppress this effect, we captured successive auto-exposure shots of each scene (5 shots for Train-Def/Var and 4 shots for Test-Def/Var), each preceded by a different gain initialization, and average them to obtain a stable reference image.

**Ground-Truth annotation.** For each pose, we used the 50%-brightness, averaged auto-exposure RGB frame and went through the annotation process below, resulting in about 2 mm errors.

- **Object masks:** Initially, rough bounding boxes were generated using Grounding DINO (Liu et al., 2024), guided by shape-aware textual prompts tailored to each object. Boxes were manually reviewed and corrected if needed. The refined boxes were passed to FastSAM (Zhao et al., 2023) to generate precise instance-level masks.
- **Object poses:** An initial camera-to-object transformation was provided by detecting the ChArUco markerboard rigidly attached to the turntable. The object poses were then refined precisely in 3D using the annotation utility (Gouda et al., 2022) in the BOP toolkit (Hodan & Sundermeyer, 2020) with depth maps captured in High Density mode. Finally, all refined poses were re-projected onto the RGB images and visually inspected to confirm alignment accuracy.
- **Accuracy of GT annotation:** Following the protocol in (Hodan et al., 2017; Kaskman et al., 2019), we calculate the difference $\delta = d_c - d_r$, where $d_c$ denotes the captured depth map and $d_r$ denotes the rendered depth map using the GT pose. Differences over 5 cm are treated as outliers and omitted. Table 3 reports the mean $\mu_\delta$, standard deviation $\sigma_\delta$, and the mean and median of the absolute differences $\mu_{|\delta|}$ and $med_{|\delta|}$. High-accuracy depth sensing is employed for reliable evaluation of GT accuracy analysis.

## 4 EXPERIMENTS

**Models.** To quantify the effect of illumination and sensor variations—and the gains from test-time sensor control—relative to conventional training–inference paradigms (large-scale pretraining and standard supervised pipelines), we evaluate several state-of-the-art methods:

- **Pretrained, generalizable models (unseen objects; RGB-D): GigaPose** (Nguyen et al., 2024), **SAM-6D** (Lin et al., 2024a), and **FoundationPose** (Wen et al., 2024). GigaPose and SAM-6D were pretrained on the large-scale MegaPose (Labbé et al., 2022), which contains objects from ShapeNet (Chang et al., 2015) and GSO (Downs et al., 2022). FoundationPose uses a custom data-generation pipeline with 3D assets from GSO and Objaverse (Deitke et al., 2023).

- **Instance-level models (known objects): ZebraPose** (Su et al., 2022), an RGB-based method with a ResNet-34 backbone; **GDRNPP** (Liu et al., 2025), which builds on GDR-Net (Wang et al., 2021) using a ConvNeXt (Liu et al., 2022) backbone and depth-based refinement module; and **HiPose** (Lin et al., 2024b), an RGB-D approach that fuses ConvNeXt-based RGB features with RandLA-Net (Hu et al., 2020) depth features. These models are trained on *SenseShift6D*.

**Sensor control.** We consider five regimes to evaluate the impact of sensor control: (1) **Baseline:** default settings for both RGB (e.g., auto-exposure) and depth sensors; (2) **Depth-Only:** Oracle selection for depth presets and default auto-exposure for RGB; (3) **RGB-Only:** Oracle selection for RGB exposure and gain, with the Default mode for the depth sensor; (4) **Oracle-Fixed:** a single best-performing fixed RGB-D configuration, applied to all test scenes; (5) **Oracle-Dynamic:** per-scene Oracle selection over both RGB and depth. Here, *Oracle* represents an upper bound obtained by evaluating the full sensor-configuration grid and counting a scene as correct if *any* configuration yields a correct prediction.

**Metrics.** We use the standard Average Distance (ADD) metric (Hinterstoisser et al., 2012b), which measures the average point-wise distance between model points transformed by the predicted pose and the ground-truth pose. Based on this metric, we report two evaluation measures: **AUC@[0:0.1]**, which corresponds to the area under the ADD recall curve up to a threshold of 10% of the object diameter, and **AR@5**, which denotes the recall at a threshold of 5% of the object diameter. Due to space limits, additional metrics—average recall of VSD, MSSD and MSPD (Hodan et al., 2018a)—are reported in Appendix D.

Table 4: **AUC@[0:0.1] performance of various sensor control methods for unseen object pose estimation models: GigaPose, SAM-6D, FoundationPose.**

| Model | Object | Baseline RGB: Auto Depth: Default | Depth-Only RGB: Auto Depth: Oracle | RGB-Only RGB: Oracle Depth: Default | Oracle-Fixed Best Fixed Param. | Oracle-Dynamic RGB: Oracle Depth: Oracle |
|---|---|---|---|---|---|---|
| GigaPose (Nguyen et al., 2024) | Spray | 63.94 | 72.41 (+8.47) | 76.42 (+12.48) | 67.98 (+4.04) | 81.24 (+17.30) |
| | Pringles | 20.91 | 46.74 (+25.83) | 46.13 (+25.22) | 30.82 (+9.91) | 70.54 (+49.63) |
| | Tincase | 64.49 | 72.67 (+8.18) | 69.32 (+4.83) | 65.51 (+1.02) | 76.41 (+11.92) |
| | Sandwich | 64.94 | 71.92 (+6.98) | 76.09 (+11.15) | 67.90 (+2.96) | 83.88 (+18.94) |
| | Mouse | 56.62 | 67.40 (+10.78) | 67.02 (+10.40) | 53.46 (-3.16) | 76.71 (+20.09) |
| | Duck | 72.26 | 78.88 (+6.62) | 77.63 (+5.37) | 71.80 (-0.46) | 82.64 (+10.38) |
| | Overall | 57.19 | 68.34 (+11.15) | 68.77 (+11.58) | 59.58 (+2.39) | 78.57 (+21.38) |
| SAM-6D (Lin et al., 2024a) | Spray | 79.49 | 89.41 (+9.92) | 87.73 (+8.24) | 81.22 (+1.73) | 90.80 (+11.31) |
| | Pringles | 63.48 | 79.17 (+15.69) | 80.86 (+17.38) | 64.54 (+1.06) | 83.98 (+20.50) |
| | Tincase | 72.26 | 83.02 (+10.76) | 85.49 (+13.23) | 68.85 (-3.41) | 86.53 (+14.27) |
| | Sandwich | 78.39 | 80.85 (+2.46) | 81.59 (+3.20) | 78.38 (-0.01) | 83.03 (+4.64) |
| | Mouse | 53.82 | 68.24 (+14.42) | 74.63 (+20.81) | 57.56 (+3.74) | 77.61 (+23.79) |
| | Duck | 76.67 | 86.21 (+9.53) | 87.35 (+10.67) | 78.58 (+1.90) | 90.70 (+14.02) |
| | Overall | 70.69 | 81.15 (+10.46) | 82.94 (+12.25) | 71.52 (+0.83) | 85.44 (+14.75) |
| FoundationPose (Wen et al., 2024) | Spray | 84.63 | 86.52 (+1.89) | 87.60 (+2.97) | 83.96 (-0.67) | 89.01 (+4.38) |
| | Pringles | 34.26 | 50.72 (+16.46) | 50.78 (+16.52) | 37.73 (+3.47) | 74.69 (+40.43) |
| | Tincase | 39.55 | 58.41 (+18.86) | 70.03 (+30.48) | 38.25 (-1.30) | 80.95 (+41.40) |
| | Sandwich | 83.17 | 86.40 (+3.23) | 85.12 (+1.95) | 83.14 (-0.03) | 88.11 (+4.94) |
| | Mouse | 70.22 | 74.34 (+4.12) | 75.86 (+5.64) | 70.69 (+0.47) | 79.98 (+9.76) |
| | Duck | 84.78 | 86.87 (+2.09) | 89.15 (+4.37) | 84.72 (-0.06) | 90.67 (+5.89) |
| | Overall | 66.10 | 73.88 (+7.78) | 76.42 (+10.32) | 66.42 (+0.32) | 83.90 (+17.80) |

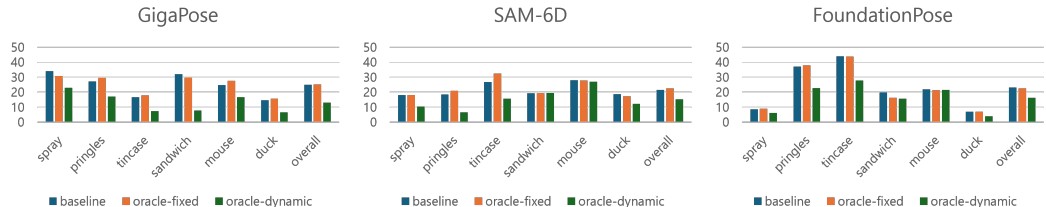

Figure 5: **Overall standard deviation of AUC computed across all brightness and scenes evaluated under Baseline, Oracle-Fixed, Oracle-Dynamic.**

## 4.1 ANALYSIS ON PRETRAINED, GENERALIZABLE POSE ESTIMATION MODELS

Generalizable 6D pose estimators are designed to cope with clutter, occlusion, and novel objects by leveraging large-scale pretraining. *SenseShift6D* removes these scene-level confounders—single object, clean background, no occlusion—which isolates the effect of environmental and sensor shifts. Evaluating GigaPose, SAM-6D, and FoundationPose on *SenseShift6D* thus tests whether robustness acquired through pretraining transfers to an *orthogonal and underexplored axis*: variations in illumination and RGB/depth sensor parameters that are largely absent from existing benchmarks.

Table 4 presents the impact of illumination and sensor variations on 6D pose estimation, evaluated with ground-truth object segmentations. Given that GigaPose and FoundationPose assume ground-truth masks, while SAM-6D includes its own image segmentation model (ISM), we also provide the results of SAM-6D's full pipeline in Appendix C for completeness.

**Default, Fixed, and Adaptive Multimodal Sensor Configurations.** Baseline (auto-exposure RGB + Default depth) yields the lowest accuracy among the five regimes. Controlling a single modality *already* helps: RGB-Only improves by +11.4 pp or more across all models, indicating that factory auto-exposure—tuned for human perception—is not model-optimal. Depth-Only also improves performance by +9.8 pp or more, showing that the default depth preset is likewise suboptimal. Furthermore, Oracle-Dynamic (per-scene RGB-and-depth selection) achieves the best results, with +18.0 pp over Baseline and +6.6 pp over the second-best RGB-Only, demonstrating complementary cues and *synergy when both sensors are adapted jointly*. In contrast, Oracle-Fixed (a single global configuration) provides only +1.2 pp on average—and can even underperform Baseline (e.g., SAM-6D on Tincase)—underscoring that *adaptive, per-scene* control is required. Crucially, all gains are obtained *without any model retraining*, highlighting the practical potential of model-specific sensor

adaptation at test time. These results motivate deeper study of **multimodal test-time sensor control** and of which RGB-D combinations most benefit each model.

**Impact on Generalization and Stability.** Under Baseline, we observe substantial *object-wise disparity*: the average gap between the best- and worst-performing objects is 42.5 pp. Such gaps arise when objects lack distinctive cues for pose disambiguation or deviate geometrically from pretraining distributions, indicating that current generalizable methods still struggle on some novel objects. In contrast, Oracle-Dynamic markedly reduces this disparity to 14.2 pp. Moreover, Figure 5 shows that per-scene sensor control reduces *scene-level variability* under the same object. The overall AUC standard deviation decreases by 8.3 pp relative to Baseline, with GigaPose on the Sandwich showing the largest reduction of 24.2 pp. This indicates higher accuracy and greater stability across diverse scenes.

Together, these findings demonstrate that adapting inputs via sensor control enhances robustness precisely where large-scale pretrained models tend to fail. While prior advances have emphasized clutter, occlusion, and category novelty, our results reveal that environmental and sensor-setting variation constitute a distinct—and underexplored—axis of difficulty warranting dedicated investigation. Qualitative examples of pretrained, generalizable models are presented in Appendix F.1.

### 4.2 ANALYSIS ON INSTANCE-LEVEL POSE ESTIMATION MODELS

Instance-level methods follow the standard train–test setup in which the training and test sets share the same object domain. Building on Section 4.1, our goal here is twofold: (i) to test whether environmental and sensor variations remain challenging once the unseen-object factor is removed, and (ii) to assess whether test-time sensor control remains effective even when the training data already includes various augmentations or explicit physical sensor and illumination variation.

For training, we generated 10k photorealistic (PBR) synthetic images per object using Blender-Proc (Denninger et al., 2023). Each model was trained independently on a per-object basis under three schemes: (1) **PBR + Train-Def** (default physical images only), (2) **PBR + Train-Def with augmentation** (including Gaussian blur, linear contrast adjustment, brightness enhancement and others; see Appendix A), and (3) **PBR + Train-Var** (full sensor and lighting variation). In all experiments, we excluded refinement modules for ZebraPose and GDRNPP, and used the full pipeline for HiPose. Since HiPose is the only RGB-D method (ZebraPose and GDRNPP are RGB-only), we report results in two parts. Table 5 evaluates the impact of illumination and RGB sensor variations (with the default depth preset) under three sensor-control settings: **Auto** (auto-exposure), **Rand** (average performance across randomly selected exposure-gain pairs), and **Oracle** (per-scene upper bound over the full RGB grid). Table 6 evaluates HiPose (training scheme 1) on Test-Var under the multimodal regimes introduced in Section 4.1.

**Sensor and Environmental Shift.** In Table 5, when trained with Train-Def, Auto performance is consistently lower on Test-Var than Test-Def, reflecting sensitivity to illumination shifts. In contrast, Oracle on Test-Var not only surpasses Auto on Test-Var but often exceeds Auto on Test-Def, showing that adapting RGB sensor settings at test time both mitigates domain shift and unlocks higher accuracy. Table 6 (RGB-D) reveals that while all sensor-control methods improve over Baseline, depth-only control provides marginal gains relative to RGB control—unlike the trend in Table 4. This is expected because: (i) instance-level training for HiPose exposes the model to the exact objects, yielding object-specific optimization that leaves less head-room for improved depth measurements; (ii) training already uses depth maps from the same domain as Baseline; and (iii) depth sensing is inherently less sensitive to illumination changes than RGB. Even with identical objects and backgrounds, however, performance still varies under environmental and sensor changes.

**Test-time Sensor Control vs. Traditional Generalization.** In Table 5, data augmentation generally improves Auto on Test-Var, but sometimes hurts , indicating that augmentation alone is not a reliable robustness solution. Training on expanded physical data (PBR + Train-Var, 3.5× larger than PBR + Train-Def) yields larger gains than augmentation, underscoring *limits of synthetic simulation in reproducing real sensor/lighting variation*. In some cases (e.g., Mouse for ZebraPose), expanded data does not improve performance on Test-Var. This is largely because severe illumination and sensor variations degrade or lose RGB cues, forcing the model to learn from missing or corrupted information, which makes it struggle to acquire stable object-specific features. Notably, Oracle with Train-Def (no augmentation) generally surpasses Auto with augmentation, and often outperforms Auto trained on Train-Var, as it selects sensor configurations that preserve valid cues at capture time,

Table 5: **AR@5 performance of various sensor control methods under different training and test settings of *SenseShift6D* for instance-level pose estimation models: ZebraPose, GDRNPP, HiPose.**

| | | ZebraPose (Su et al., 2022) | | | | GDRNPP (Liu et al., 2025) | | | | HiPose (Lin et al., 2024b) | | | |
| | | Test-Def | Test-Var | | | Test-Def | Test-Var | | | Test-Def | Test-Var | | |
| Object | Train | Auto | Auto | Rand | Oracle | Auto | Auto | Rand | Oracle | Auto | Auto | Rand | Oracle |
|---|---|---|---|---|---|---|---|---|---|---|---|---|---|
| Spray | PBR + Train-Def | 100.0 | 96.09 | 52.26 | 99.51 | 100.0 | 96.59 | 52.02 | 99.02 | 100.0 | 98.73 | 59.98 | 100.0 |
| | PBR + Train-Def w/ Aug | 100.0 | 98.04 | 68.46 | 100.0 | 100.0 | 98.54 | 71.88 | 100.0 | 100.0 | 100.0 | 89.61 | 100.0 |
| | PBR + Train-Var | 97.56 | 97.07 | 78.80 | 97.56 | 97.56 | 97.56 | 85.93 | 98.54 | 100.0 | 100.0 | 100.0 | 100.0 |
| Pringles | PBR + Train-Def | 85.71 | 75.71 | 38.87 | 94.75 | 69.05 | 60.95 | 35.64 | 89.05 | 100.0 | 96.00 | 58.74 | 100.0 |
| | PBR + Train-Def w/ Aug | 73.80 | 66.18 | 40.92 | 89.99 | 80.95 | 79.05 | 49.17 | 89.52 | 97.62 | 97.81 | 72.81 | 100.0 |
| | PBR + Train-Var | 90.47 | 89.04 | 68.35 | 95.23 | 90.48 | 90.00 | 72.22 | 97.14 | 97.62 | 97.62 | 86.86 | 98.10 |
| Tincase | PBR + Train-Def | 95.56 | 79.56 | 52.53 | 96.44 | 91.11 | 81.33 | 46.58 | 92.89 | 97.78 | 81.51 | 53.62 | 96.44 |
| | PBR + Train-Def w/ Aug | 97.78 | 89.78 | 65.46 | 100.0 | 80.00 | 82.22 | 52.89 | 95.11 | 97.78 | 95.64 | 86.47 | 97.78 |
| | PBR + Train-Var | 98.22 | 97.60 | 79.59 | 100.0 | 100.0 | 99.56 | 82.02 | 100.0 | 97.78 | 97.69 | 97.76 | 97.78 |
| Sandwich | PBR + Train-Def | 94.87 | 89.23 | 48.26 | 100.0 | 97.44 | 88.21 | 47.03 | 96.41 | 94.87 | 87.69 | 57.21 | 94.87 |
| | PBR + Train-Def w/ Aug | 92.31 | 88.72 | 57.82 | 98.97 | 94.87 | 92.31 | 63.72 | 98.97 | 94.87 | 93.74 | 84.38 | 94.87 |
| | PBR + Train-Var | 89.74 | 89.23 | 67.15 | 96.93 | 97.44 | 97.44 | 82.64 | 100.0 | 94.87 | 94.87 | 94.87 | 94.87 |
| Mouse | PBR + Train-Def | 70.73 | 61.46 | 27.37 | 86.34 | 95.12 | 76.10 | 31.44 | 88.78 | 90.24 | 62.93 | 43.73 | 87.81 |
| | PBR + Train-Def w/ Aug | 48.78 | 49.76 | 32.56 | 88.29 | 78.05 | 72.68 | 43.29 | 86.83 | 90.24 | 89.07 | 82.24 | 90.24 |
| | PBR + Train-Var | 46.34 | 46.34 | 34.98 | 73.17 | 80.49 | 80.49 | 66.39 | 91.22 | 90.24 | 89.95 | 87.17 | 90.73 |
| Duck | PBR + Train-Def | 69.77 | 83.26 | 34.08 | 95.81 | 95.35 | 84.65 | 39.63 | 99.07 | 97.67 | 75.16 | 44.26 | 99.54 |
| | PBR + Train-Def w/ Aug | 86.05 | 84.19 | 59.00 | 99.07 | 90.70 | 84.65 | 57.91 | 99.07 | 97.67 | 99.26 | 82.72 | 100.0 |
| | PBR + Train-Var | 89.30 | 91.63 | 74.49 | 99.07 | 97.67 | 97.67 | 81.42 | 99.53 | 97.67 | 98.70 | 97.37 | 100.0 |
| Overall | PBR + Train-Def | 86.11 | 80.89 | 42.23 | 95.47 | 91.35 | 81.31 | 42.06 | 94.20 | 96.76 | 83.67 | 52.92 | 96.44 |
| | PBR + Train-Def w/ Aug | 83.12 | 79.44 | 54.04 | 96.05 | 87.43 | 84.91 | 56.48 | 94.92 | 96.36 | 95.92 | 83.04 | 97.15 |
| | PBR + Train-Var | 85.27 | 85.15 | 67.23 | 93.66 | 93.94 | 93.79 | 78.44 | 97.74 | 96.36 | 96.47 | 94.01 | 96.91 |

Table 6: **AUC@[0:0.1] performance of various sensor control methods for HiPose, trained on PBR + Train-Def and tested on Test-Var.**

| Object | Baseline
RGB: Auto
Depth: Default | Depth-Only
RGB: Auto
Depth: Oracle | RGB-Only
RGB: Oracle
Depth: Default | Oracle-Fixed
Best Fixed Param. | Oracle-Dynamic
RGB: Oracle
Depth: Oracle |
|---|---|---|---|---|---|
| Spray | 86.92 | 88.17 (+1.25) | 88.79 (+1.87) | 87.54 (+0.62) | 89.87 (+2.95) |
| Pringles | 73.40 | 76.18 (+2.78) | 79.08 (+5.68) | 77.02 (+3.62) | 81.55 (+8.15) |
| Tincase | 74.78 | 77.88 (+3.10) | 85.30 (+10.52) | 83.57 (+8.79) | 86.84 (+12.06) |
| Sandwich | 78.68 | 81.72 (+3.04) | 84.06 (+5.38) | 82.90 (+4.22) | 86.43 (+7.75) |
| Mouse | 63.92 | 66.41 (+2.49) | 72.19 (+8.27) | 69.52 (+5.60) | 75.21 (+11.29) |
| Duck | 70.27 | 72.94 (+2.67) | 86.67 (+16.40) | 65.54 (-4.73) | 88.46 (+18.19) |
| Overall | 74.66 | 77.22 (+2.44) | 82.68 (+8.02) | 77.68 (+3.02) | 84.73 (+ 10.07) |

allowing the downstream pose estimator to operate on more reliable observations. For example, Oracle trained on Train-Def achieves over 10% higher accuracy than Auto trained on Train-Var for ZebraPose.

These results demonstrate the substantial potential of test-time sensor control relative to conventional generalization strategies (augmentation or scaling training data), *even for known objects*. Mitigating covariate shift at inference via sensor optimization can be more effective—or complementary—than exclusively attempting to generalize the model to all possible shifts. Qualitative results are provided in Appendix F.2.

## 5 CONCLUSION

We introduced *SenseShift6D*, a *physically* captured RGB-D benchmark that orthogonally sweeps exposure, gain, depth-capture mode, and illumination to isolate the impact of sensor and environmental shifts on 6D object pose estimation. Across pretrained, generalizable models (GigaPose, SAM-6D, FoundationPose) and instance-level models (ZebraPose, GDRNPP, HiPose), we found that factory defaults are not model-optimal: RGB-Only control improves accuracy by +11.4 pp or more and Depth-Only by +9.8 pp or more, while multimodal (RGB+Depth) control achieves the best results (+18.0 pp over Baseline; +6.6 pp over RGB-Only). A single global configuration (Oracle-Fixed) yields only +1.2 pp on average (and can underperform Baseline), underscoring the need for *adaptive, per-scene* control. Beyond mean accuracy, per-scene control reduces object-wise disparity from 42.5 pp to 14.2 pp and lowers scene-level variability by 8.3 pp, indicating improved *stability* as well as

accuracy. Crucially, these gains require *no retraining*, positioning test-time sensor control as a data- and compute-efficient complement to model-centric scaling.

**Limitations and Future Work.** *SenseShift6D* currently focuses on single-object tabletop scenes and varies only brightness intensity, omitting complexities such as directional lighting, shadows, and occlusions. Future extensions of the dataset will include: (1) multi-object scenarios to explore occlusion and clutter; (2) directional and shadow-rich lighting conditions; and (3) a real-time adaptive sensor-control framework that dynamically adjusts RGB and depth parameters. We believe *SenseShift6D* provides a strong foundation for developing robust, sensor-adaptive 6D pose estimation systems suitable for deployment in diverse and dynamic real-world environments.

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

# APPENDIX

## A  RGB AUGMENTATION SETTINGS

To improve generalization under varying environmental conditions, we adopted an RGB augmentation pipeline based on that used in GDRNPP. The augmentations were implemented using the `imgaug` library and applied only to the RGB input during training with a probability of 0.8. An overview of these augmentations is illustrated in Figure 6.

The augmentations listed in Table 7 were applied in random order, with each operation activated independently with a fixed probability.

Table 7: **Augmentation operations and parameters.**

| Augmentation Type | Activation Probability | Parameters / Range |
|---|---|---|
| Gaussian Blur | 0.4 | $\sigma \in [0.0, 3.0]$ |
| Sharpness Enhancement | 0.3 | factor $\in [0.0, 50.0]$ |
| Contrast Enhancement | 0.3 | factor $\in [0.2, 50.0]$ |
| Brightness Enhancement | 0.5 | factor $\in [0.1, 6.0]$ |
| Color Enhancement | 0.3 | factor $\in [0.0, 20.0]$ |
| Additive Intensity | 0.5 | value $\in [-25, 25]$, per-channel = 0.3 |
| Invert Pixels | 0.3 | probability = 0.2, per-channel |
| Multiply Intensity | 0.5 | factor $\in [0.6, 1.4]$, per-channel = 0.5 |
| Additive Gaussian Noise | 0.1 | scale = 10, per-channel |
| Linear Contrast | 0.5 | factor $\in [0.5, 2.2]$, per-channel = 0.3 |
| Grayscale Conversion | 0.5 | $\alpha \in [0.0, 1.0]$ |

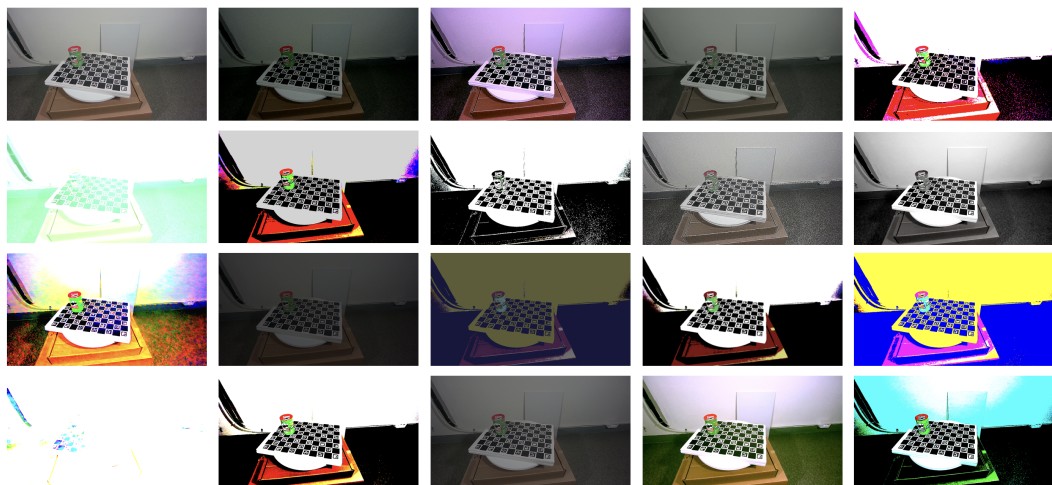

Figure 6: **Examples of RGB augmentations applied during training.** The top-left image is the original, and the remaining images are randomly augmented samples generated using the settings summarized in Table 7.

## B  EXPERIMENTAL DETAILS

### B.1  COMPUTING RESOURCES

All models were trained on a high-performance computing cluster equipped with NVIDIA GPUs and Intel Xeon CPUs. Specifically, *ZebraPose* and *GDRNPP* were trained on nodes with RTX 3090 GPUs (24GB VRAM) and Intel Xeon Silver 4210R CPUs, while *HiPose* was trained on a node with

A100 GPUs (80GB VRAM) and an Intel Xeon Gold 6530 CPU. Each training node had access to 512GB of RAM.

### B.2 Pretrained Weights for Unseen Object Pose Estimation Models

For GigaPose, we used the pretrained weight files 'gigaPose_v1' (coarse) and 'refiner-rgbd-288182519' (refiner), following instructions from GigaPose GitHub repository.

For FoundationPose, we used the pretrained weight files '2023-10-28-18-33-37' (refiner) and '2024-01-11-20-02-45' (scorer), following instructions from FoundationPose GitHub repository.

For SAM-6D, we used the pretrained weight file 'vit_h' for SAM and the 'sam-6d-pem-base.pth' file for the Pose Estimation Module (PEM), following instructions from SAM-6D Github repository.

### B.3 Model Training for Instance-Level Pose Estimation Models

All models were trained with a batch size of 32. *ZebraPose* and *HiPose* were both trained for a total of 38,000 iterations using the Adam optimizer, with learning rates of 0.0002 and 0.0001, respectively. *GDRNPP* was trained for 120 epochs using the Ranger optimizer with a learning rate of 8e-4 and weight decay of 0.01. A flat-and-anneal learning rate schedule with cosine annealing was applied, starting at 72% of the total training epochs. A linear warmup was used for the first 1,000 iterations with a warmup factor of 0.001. All learning rate schedules and optimizer settings follow the original implementations of each method.

## C Evaluation of the full SAM-6D pipeline

Table 8 presents SAM-6D performance based on the Top-30 segmentation results produced by its ISM.

Table 8: **AUC@[0:0.1] performance of SAM-6D with its Image Segmentation Model (ISM).**

| Object | Baseline RGB: Auto Depth: Default | Depth-Only RGB: Auto Depth: Oracle | RGB-Only RGB: Oracle Depth: Default | Oracle-Fixed Best Fixed Param. | Oracle-Dynamic RGB: Oracle Depth: Oracle |
|---|---|---|---|---|---|
| Spray | 81.17 | 88.93 (+7.76) | 89.88 (+8.71) | 78.71 (-2.46) | 90.66 (+9.49) |
| Pringles | 61.84 | 78.71 (+16.87) | 79.76 (+17.92) | 55.85 (-5.99) | 83.14 (+21.30) |
| Tincase | 63.85 | 82.13 (+18.28) | 82.16 (+18.31) | 60.02 (-3.83) | 85.42 (+21.57) |
| Sandwich | 74.66 | 81.13 (+6.47) | 81.62 (+6.96) | 71.47 (-3.19) | 82.82 (+8.16) |
| Mouse | 49.16 | 66.49 (+17.33) | 71.46 (+22.30) | 47.96 (-1.20) | 74.37 (+25.21) |
| Overall | 66.13 | 79.48 (+13.35) | 80.97 (+14.84) | 62.80 (-3.33) | 83.28 (+17.15) |

## D Evaluation on BOP metrics

Table 9 reports the evaluation of unseen object pose estimation models—GigaPose, SAM-6D, and FoundationPose—using the BOP metrics: VSD, MSSD, and MSPD.

Table 9: **Average Recall of VSD, MSSD, and MSPD on various sensor control methods for unseen object pose estimation models: GigaPose, SAM-6D, FoundationPose.**

| Model | Object | Baseline
RGB: Auto
Depth: Default | Depth-Only
RGB: Auto
Depth: Oracle | RGB-Only
RGB: Oracle
Depth: Default | Oracle-Fixed
Best Fixed Param. | Oracle-Dynamic
RGB: Oracle
Depth: Oracle |
|---|---|---|---|---|---|---|
| GigaPose
(Nguyen et al., 2024) | Spray | 71.42 / 82.68 / 84.15 | 84.46 / 88.68 / 90.39 | 83.19 / 94.05 / 94.54 | 77.76 / 86.15 / 87.71 | 91.44 / 95.22 / 96.20 |
| | Pringles | 79.30 / 68.24 / 84.29 | 88.88 / 83.14 / 95.33 | 93.55 / 87.90 / 95.10 | 84.03 / 75.76 / 90.00 | 97.72 / 96.05 / 99.43 |
| | Tincase | 85.04 / 95.87 / 99.20 | 90.87 / 99.16 / 99.96 | 90.72 / 97.16 / 99.60 | 86.87 / 94.93 / 97.38 | 96.48 / 99.60 / 100.0 |
| | Sandwich | 80.50 / 83.28 / 86.15 | 88.76 / 87.54 / 87.69 | 91.89 / 96.31 / 99.18 | 85.12 / 85.64 / 86.36 | 98.21 / 99.95 / 100.0 |
| | Mouse | 80.26 / 84.44 / 94.34 | 88.41 / 91.02 / 96.44 | 86.94 / 91.90 / 98.34 | 79.34 / 79.76 / 88.34 | 93.90 / 96.78 / 99.61 |
| | Overall | 79.31 / 82.90 / 89.63 | 88.28 / 89.91 / 93.96 | 89.26 / 93.46 / 97.35 | 82.62 / 84.45 / 89.96 | 95.55 / 97.52 / 99.05 |
| SAM-6D
(Lin et al., 2024a) | Spray | 92.47 / 96.33 / 96.53 | 94.96 / 98.00 / 98.15 | 91.00 / 93.66 / 93.66 | 92.77 / 96.33 / 96.53 | 97.43 / 99.22 / 99.17 |
| | Pringles | 94.38 / 93.08 / 98.56 | 97.05 / 97.76 / 100.0 | 95.96 / 96.57 / 98.57 | 94.32 / 92.43 / 96.81 | 99.16 / 99.62 / 100.0 |
| | Tincase | 97.53 / 88.95 / 90.32 | 96.47 / 93.33 / 94.36 | 96.00 / 94.67 / 95.16 | 97.31 / 82.15 / 83.52 | 97.33 / 97.20 / 97.82 |
| | Sandwich | 88.74 / 94.87 / 94.87 | 89.42 / 94.87 / 94.92 | 90.19 / 94.87 / 95.38 | 88.65 / 94.87 / 94.87 | 91.32 / 95.44 / 96.31 |
| | Mouse | 78.79 / 79.49 / 85.10 | 82.08 / 84.41 / 89.12 | 84.92 / 87.06 / 91.42 | 79.04 / 81.56 / 86.87 | 88.56 / 89.85 / 93.92 |
| | Overall | 90.38 / 90.54 / 93.08 | 92.00 / 93.68 / 95.31 | 91.62 / 93.37 / 94.84 | 90.42 / 89.47 / 91.72 | 94.76 / 96.27 / 97.44 |
| FoundationPose
(Wen et al., 2024) | Spray | 89.45 / 99.22 / 99.32 | 92.85 / 99.76 / 99.85 | 94.27 / 100.0 / 100.0 | 89.49 / 99.32 / 99.37 | 97.15 / 100.0 / 100.0 |
| | Pringles | 93.98 / 58.29 / 75.29 | 96.16 / 77.00 / 86.38 | 96.41 / 81.10 / 89.52 | 94.14 / 62.43 / 77.38 | 99.12 / 96.10 / 98.33 |
| | Tincase | 97.36 / 44.89 / 49.11 | 97.60 / 65.64 / 68.09 | 97.62 / 79.11 / 81.73 | 96.96 / 43.56 / 47.33 | 97.75 / 89.69 / 92.09 |
| | Sandwich | 94.49 / 95.08 / 96.31 | 97.26 / 97.59 / 98.97 | 95.75 / 96.26 / 98.26 | 95.87 / 97.38 / 97.85 | 97.96 / 98.41 / 99.49 |
| | Mouse | 88.39 / 90.59 / 94.73 | 91.24 / 92.59 / 97.66 | 90.72 / 91.95 / 96.68 | 89.02 / 91.56 / 96.54 | 93.66 / 94.15 / 98.44 |
| | Overall | 92.73 / 77.61 / 82.95 | 95.02 / 86.52 / 90.19 | 94.95 / 89.68 / 93.24 | 93.10 / 78.85 / 83.69 | 97.13 / 95.67 / 97.67 |

# E  EVALUATION ON MARKERLESS TEST SCENES

Since visible markers in the scene can introduce model bias (Govi et al., 2023), we modified Test-Var split images (see Figure 7) and evaluated them using the same model weights as in Section 4.2. In Table 10, the RGB-D based model (i.e., HiPose (Lin et al., 2024b)), maintains performance comparable to Table 5, whereas the RGB-based models (i.e., ZebraPose (Su et al., 2022) and GDRNPP (Liu et al., 2025)) show substantial object-dependent variation. Both RGB models retain similar performance for Spray compared to Table 5, but exhibit a dramatic drop for Mouse. This indicates that Spray, which has rich RGB features, can be learned without relying on markers, while Mouse, which has ambiguous shape and pale coloration, lacks discriminative features and thus the models had relied heavily on markers during training. Despite this discrepancy, sensor optimization consistently adapts to environmental changes more effectively than existing generalization techniques, even without additional training across all objects.

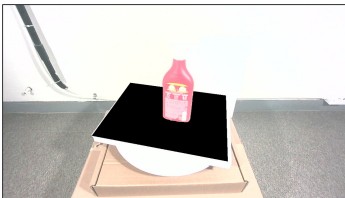 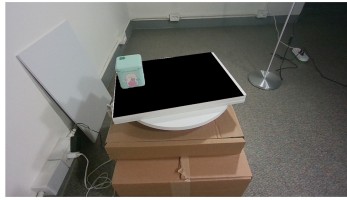 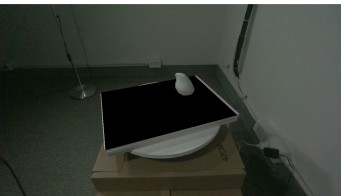

Figure 7: **Sample of marker-removed test images.**

Table 10: **AR@5 comparison of sensor control methods on marker-removed *SenseShift6D* with ZebraPose, GDRNPP, and HiPose.**

| | | ZebraPose (Su et al., 2022) | | | | GDRNPP (Liu et al., 2025) | | | | HiPose (Lin et al., 2024b) | | | |
|---|---|---|---|---|---|---|---|---|---|---|---|---|---|
| | | Test-Def | Test-Var | | | Test-Def | Test-Var | | | Test-Def | Test-Var | | |
| Object | Train | Auto | Auto | Rand | Oracle | Auto | Auto | Rand | Oracle | Auto | Auto | Rand | Oracle |
| Spray | PBR + Train-Def | 96.59 | 95.12 | 64.33 | 100 | 100 | 96.59 | 52.49 | 99.51 | 100 | 98.73 | 59.98 | 100 |
| | PBR + Train-Def w/ Aug | 93.66 | 93.17 | 70.71 | 100 | 95.12 | 94.63 | 71.63 | 100 | 100 | 100 | 91.66 | 100 |
| | PBR + Train-Var | 98.05 | 97.07 | 78.91 | 100 | 100 | 100 | 81.89 | 100 | 100 | 100 | 100 | 100 |
| Tincase | PBR + Train-Def | 94.22 | 86.22 | 55.45 | 99.11 | 64.44 | 58.66 | 35.07 | 84.89 | 97.78 | 84.62 | 56.33 | 97.78 |
| | PBR + Train-Def w/ Aug | 76.44 | 72.89 | 52.62 | 87.11 | 40 | 38.22 | 26.69 | 61.33 | 97.78 | 96.89 | 88.40 | 97.78 |
| | PBR + Train-Var | 85.33 | 81.33 | 59.79 | 96.00 | 51.11 | 48.44 | 34.40 | 71.56 | 97.78 | 97.69 | 97.73 | 97.78 |
| Mouse | PBR + Train-Def | 48.78 | 43.41 | 30.63 | 84.88 | 4.88 | 3.90 | 2.98 | 29.27 | 90.24 | 67.42 | 47.17 | 88.78 |
| | PBR + Train-Def w/ Aug | 26.83 | 24.39 | 18.83 | 57.56 | 12.2 | 12.17 | 11.66 | 45.37 | 90.24 | 88.49 | 84.56 | 90.24 |
| | PBR + Train-Var | 9.76 | 8.29 | 5.07 | 25.37 | 2.44 | 1.95 | 3.39 | 19.02 | 90.24 | 90.15 | 87.68 | 90.73 |

# F    QUALITATIVE ANALYSIS

## F.1    QUALITATIVE RESULT ON PRETRAINED, GENERALIZABLE POSE ESTIMATION MODELS

Figure 8, 9, 10 visualize GT and predicted poses on Baseline and Oracle sensor configuration for GigaPose, SAM-6D and FoundationPose.

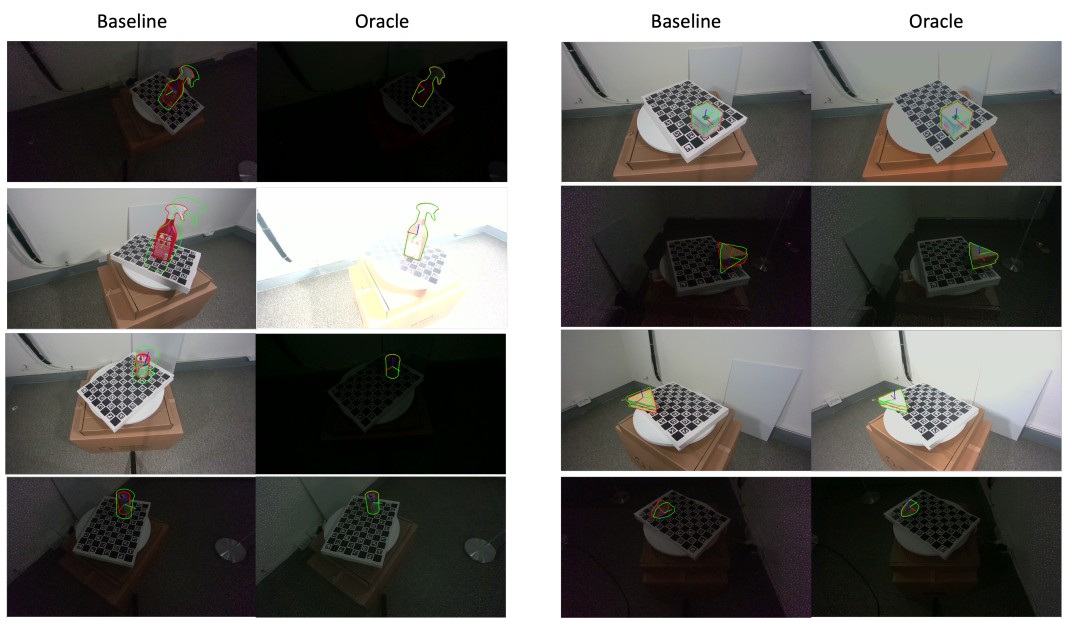

Figure 8: **Comparison of predictions under Baseline and Oracle for GigaPose.** Visualized object pose on RGB images: ground truth pose in red, predicted pose in green.

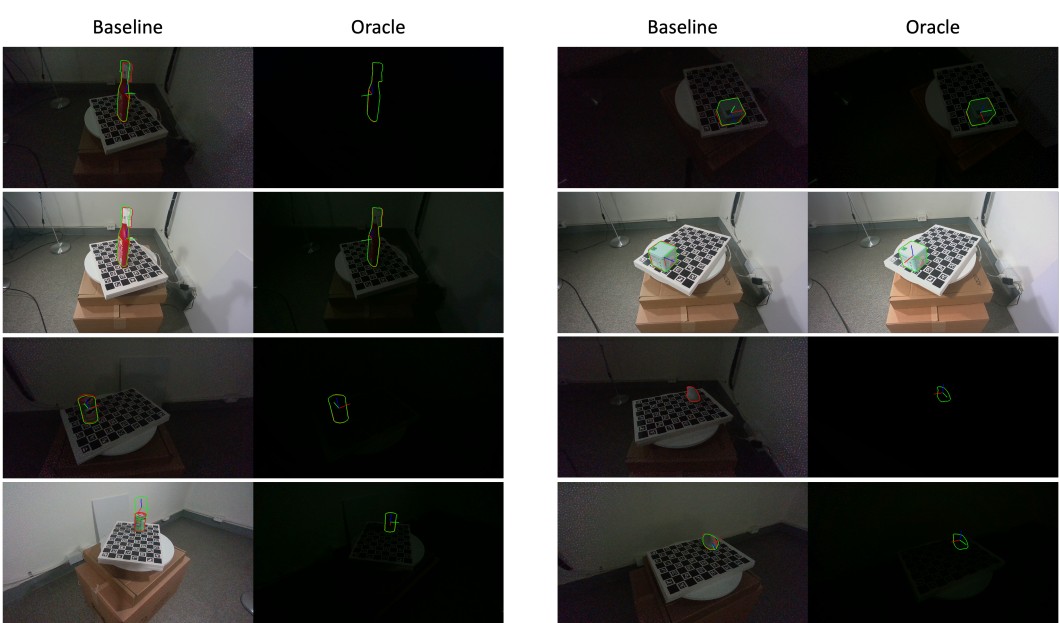

Figure 9: **Comparison of predictions under Baseline and Oracle for SAM-6D.** Visualized object pose on RGB images: ground truth pose in red, predicted pose in green.

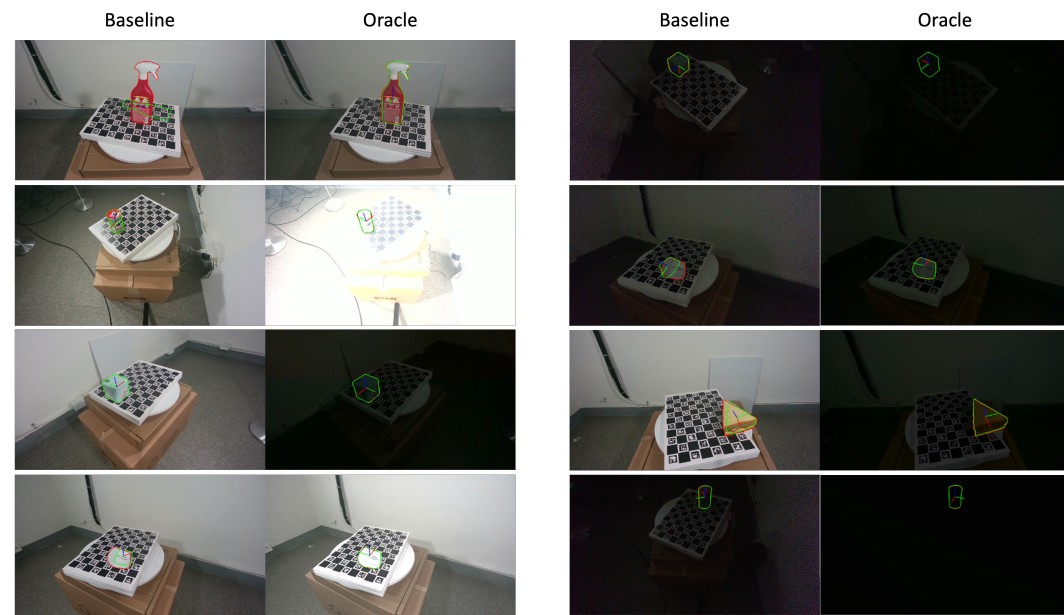

Figure 10: **Comparison of predictions under Baseline and Oracle for FoundationPose.** Visualized object pose on RGB images: ground truth pose in red, predicted pose in green.

### F.2   QUALITATIVE RESULT ON INSTANCE-LEVEL POSE ESTIMATION MODEL

Figure 11 visualizes GT and predicted poses on Baseline and Oracle sensor configuration for HiPose.

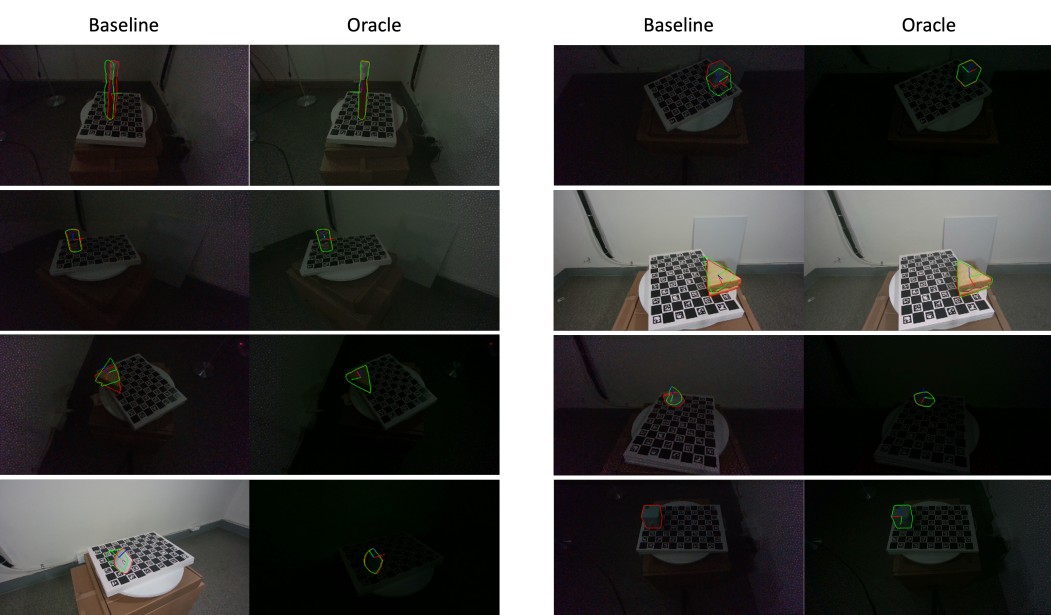

Figure 11: **Comparison of predictions under Baseline and Oracle for HiPose.** Visualized object pose on RGB images: ground truth pose in red, predicted pose in green.

# G    EVALUATION ON RGB-D BASED MODELS UNDER DIFFERENT ENVIRONMENTS

Table 11, 12, 13, 14 shows performance of RGB-D models—HiPose, GigaPose, SAM-6D, and FoundationPose—under different illumination conditions.

Table 11: **AUC@[0:0.1] performance under diverse environments of various sensor control methods for HiPose.**

| Environment | Object | Baseline RGB: Auto Depth: Default | Depth-Only RGB: Auto Depth: Oracle | RGB-Only RGB: Oracle Depth: Default | Oracle-Fixed Best Fixed Param. | Oracle-Dynamic RGB: Oracle Depth: Oracle |
|---|---|---|---|---|---|---|
| B5 | Spray | 83.17 | 85.07 (+1.90) | 87.10 (+3.93) | 86.07 (+2.90) | 88.63 (+5.46) |
| | Pringles | 72.40 | 75.45 (+3.05) | 78.60 (+6.19) | 77.90 (+5.50) | 81.12 (+8.71) |
| | Tincase | 32.96 | 43.00 (+10.04) | 78.80 (+45.84) | 76.69 (+43.73) | 81.31 (+48.36) |
| | Sandwich | 66.85 | 69.85 (+3.00) | 80.72 (+13.87) | 78.82 (+11.97) | 83.18 (+16.33) |
| | Mouse | 35.29 | 39.88 (+4.59) | 64.85 (+29.56) | 62.93 (+27.63) | 69.88 (+34.59) |
| | Overall | 58.13 | 62.65 (+4.52) | 78.01 (+19.88) | 76.48 (+18.35) | 80.82 (+22.69) |
| B25 | Spray | 87.56 | 88.71 (+1.15) | 88.61 (+1.05) | 87.59 (+0.02) | 89.63 (+2.07) |
| | Pringles | 74.12 | 77.17 (+3.05) | 79.14 (+5.02) | 77.31 (+3.19) | 81.74 (+7.62) |
| | Tincase | 85.20 | 86.33 (+1.13) | 86.91 (+1.71) | 85.13 (-0.07) | 87.89 (+2.69) |
| | Sandwich | 80.54 | 83.51 (+2.97) | 83.31 (+2.77) | 82.97 (+2.44) | 86.51 (+5.97) |
| | Mouse | 70.41 | 73.17 (+2.76) | 74.32 (+3.90) | 71.54 (+1.12) | 77.29 (+6.88) |
| | Overall | 79.57 | 81.78 (+2.21) | 82.46 (+2.89) | 80.91 (+1.34) | 84.61 (+5.04) |
| B50 | Spray | 87.93 | 88.78 (+0.85) | 89.32 (+1.39) | 88.07 (+0.15) | 90.07 (+2.15) |
| | Pringles | 74.24 | 76.57 (+2.33) | 79.69 (+5.45) | 76.74 (+2.50) | 81.62 (+7.38) |
| | Tincase | 85.11 | 86.69 (+1.58) | 86.91 (+1.80) | 85.29 (+0.18) | 88.31 (+3.20) |
| | Sandwich | 81.38 | 84.38 (+3.00) | 85.72 (+4.33) | 84.03 (+2.64) | 87.23 (+5.85) |
| | Mouse | 71.93 | 73.17 (+1.24) | 74.51 (+2.59) | 71.63 (-0.29) | 77.44 (+5.51) |
| | Overall | 80.12 | 81.92 (+1.80) | 83.23 (+3.11) | 81.15 (+1.03) | 84.93 (+4.81) |
| B75 | Spray | 88.02 | 89.02 (+1.00) | 89.51 (+1.49) | 88.02 (+0.00) | 90.61 (+2.59) |
| | Pringles | 73.31 | 76.02 (+2.71) | 79.12 (+5.81) | 76.83 (+3.52) | 81.86 (+8.55) |
| | Tincase | 85.27 | 86.49 (+1.22) | 86.80 (+1.53) | 85.33 (+0.07) | 88.13 (+2.87) |
| | Sandwich | 82.10 | 85.13 (+3.03) | 84.41 (+2.31) | 84.38 (+2.28) | 87.10 (+5.00) |
| | Mouse | 71.73 | 73.27 (+1.54) | 74.17 (+2.44) | 71.24 (-0.49) | 76.07 (+4.34) |
| | Overall | 80.09 | 81.99 (+1.90) | 82.80 (+2.71) | 81.16 (+1.07) | 84.75 (+4.66) |
| B100 | Spray | 87.93 | 89.27 (+1.34) | 89.39 (+1.46) | 87.95 (+0.02) | 90.41 (+2.49) |
| | Pringles | 72.90 | 75.69 (+2.79) | 78.86 (+5.95) | 76.31 (+3.40) | 81.40 (+8.50) |
| | Tincase | 85.38 | 86.87 (+1.49) | 87.09 (+1.71) | 85.40 (+0.02) | 88.53 (+3.16) |
| | Sandwich | 82.51 | 85.72 (+3.21) | 86.15 (+3.64) | 84.31 (+1.80) | 88.10 (+5.59) |
| | Mouse | 70.24 | 72.59 (+2.34) | 73.07 (+2.83) | 70.24 (+0.00) | 75.39 (+5.15) |
| | Overall | 79.79 | 82.03 (+2.24) | 82.91 (+3.12) | 80.84 (+1.05) | 84.77 (+4.98) |

Table 12: **AUC@[0:0.1] performance under diverse environments of various sensor control methods for GigaPose.**

| Environment | Object | Baseline RGB: Auto Depth: Default | Depth-Only RGB: Auto Depth: Oracle | RGB-Only RGB: Oracle Depth: Default | Oracle-Fixed Best Fixed Param. | Oracle-Dynamic RGB: Oracle Depth: Oracle |
|---|---|---|---|---|---|---|
| B5 | Spray | 65.76 | 73.34 (+7.58) | 73.39 (+7.63) | 66.51 (+0.75) | 78.41 (+12.65) |
| | Pringles | 19.17 | 44.19 (+25.02) | 44.64 (+25.47) | 31.33 (+12.16) | 66.69 (+47.52) |
| | Tincase | 60.69 | 71.51 (+10.82) | 65.20 (+4.51) | 65.89 (+5.20) | 75.07 (+14.38) |
| | Sandwich | 59.90 | 68.82 (+8.92) | 69.46 (+9.56) | 61.36 (+1.46) | 80.28 (+20.38) |
| | Mouse | 48.27 | 60.44 (+12.17) | 63.80 (+15.53) | 55.59 (+7.32) | 74.29 (+26.02) |
| | Overall | 50.76 | 63.66 (+12.90) | 63.30 (+12.54) | 56.14 (+5.38) | 74.95 (+24.19) |
| B25 | Spray | 67.44 | 73.20 (+5.76) | 77.78 (+10.34) | 70.49 (+3.05) | 81.15 (+13.71) |
| | Pringles | 15.62 | 49.24 (+33.62) | 49.36 (+33.74) | 38.17 (+22.55) | 73.88 (+58.26) |
| | Tincase | 63.07 | 72.13 (+9.06) | 69.29 (+6.22) | 67.29 (+4.22) | 75.69 (+12.62) |
| | Sandwich | 68.08 | 74.18 (+6.10) | 77.72 (+9.64) | 75.79 (+7.71) | 84.59 (+16.51) |
| | Mouse | 61.93 | 69.71 (+7.78) | 68.15 (+6.22) | 61.95 (+0.02) | 77.00 (+15.07) |
| | Overall | 55.23 | 67.69 (+12.46) | 68.46 (+13.23) | 62.74 (+7.51) | 78.46 (+23.23) |
| B50 | Spray | 65.51 | 73.71 (+8.20) | 77.20 (+11.69) | 71.10 (+5.59) | 83.78 (+18.27) |
| | Pringles | 24.43 | 45.40 (+20.97) | 43.33 (+18.90) | 33.50 (+9.07) | 70.14 (+45.71) |
| | Tincase | 64.98 | 72.42 (+7.44) | 70.22 (+5.24) | 68.31 (+3.33) | 76.82 (+11.84) |
| | Sandwich | 63.92 | 70.46 (+6.54) | 77.51 (+13.59) | 78.03 (+14.11) | 84.38 (+20.46) |
| | Mouse | 56.93 | 68.93 (+12.00) | 67.98 (+11.05) | 63.68 (+6.75) | 78.20 (+21.27) |
| | Overall | 55.15 | 66.18 (+11.03) | 67.25 (+12.10) | 62.92 (+7.77) | 78.67 (+23.52) |
| B75 | Spray | 61.41 | 71.10 (+9.69) | 77.37 (+15.96) | 70.68 (+9.27) | 81.41 (+20.00) |
| | Pringles | 22.88 | 47.57 (+24.69) | 45.69 (+22.81) | 35.12 (+12.24) | 72.00 (+49.12) |
| | Tincase | 66.78 | 73.36 (+6.58) | 71.49 (+4.71) | 70.78 (+4.00) | 77.36 (+10.58) |
| | Sandwich | 66.72 | 73.00 (+6.28) | 77.79 (+11.07) | 78.44 (+11.72) | 84.95 (+18.23) |
| | Mouse | 55.59 | 68.66 (+13.07) | 67.63 (+12.04) | 63.61 (+8.02) | 76.93 (+21.34) |
| | Overall | 54.68 | 66.74 (+12.06) | 67.99 (+13.31) | 63.73 (+9.05) | 78.53 (+23.85) |
| B100 | Spray | 59.59 | 70.71 (+11.12) | 76.37 (+16.78) | 70.71 (+11.12) | 81.44 (+21.85) |
| | Pringles | 22.48 | 47.31 (+24.83) | 47.64 (+25.16) | 33.12 (+10.64) | 70.00 (+47.52) |
| | Tincase | 66.93 | 73.91 (+6.98) | 70.42 (+3.49) | 69.38 (+2.45) | 77.13 (+10.20) |
| | Sandwich | 66.10 | 73.15 (+7.05) | 77.95 (+11.85) | 76.77 (+10.67) | 85.21 (+19.11) |
| | Mouse | 60.39 | 69.29 (+8.90) | 67.54 (+7.15) | 63.12 (+2.73) | 77.15 (+16.76) |
| | Overall | 55.10 | 66.87 (+11.77) | 67.98 (+12.88) | 62.62 (+7.52) | 78.18 (+23.08) |

Table 13: **AUC@[0:0.1] performance under diverse environments of various sensor control methods for SAM-6D.**

| Environment | Object | Baseline RGB: Auto Depth: Default | Depth-Only RGB: Auto Depth: Oracle | RGB-Only RGB: Oracle Depth: Default | Oracle-Fixed Best Fixed Param. | Oracle-Dynamic RGB: Oracle Depth: Oracle |
|---|---|---|---|---|---|---|
| B5 | Spray | 80.41 | 86.83 (+6.42) | 84.27 (+3.86) | 82.66 (+2.25) | 89.27 (+8.86) |
| | Pringles | 48.98 | 71.45 (+22.47) | 73.10 (+24.12) | 65.76 (+16.78) | 80.69 (+31.71) |
| | Tincase | 55.80 | 74.82 (+19.02) | 79.64 (+23.84) | 72.60 (+16.80) | 83.96 (+28.16) |
| | Sandwich | 78.36 | 79.00 (+0.64) | 80.41 (+2.05) | 78.95 (+0.59) | 81.15 (+2.79) |
| | Mouse | 49.85 | 63.24 (+13.39) | 67.61 (+17.76) | 58.07 (+8.22) | 72.90 (+23.05) |
| | Overall | 62.68 | 75.07 (+12.39) | 77.01 (+14.33) | 71.61 (+8.93) | 81.59 (+18.91) |
| B25 | Spray | 76.00 | 85.90 (+9.90) | 77.88 (+1.88) | 82.29 (+6.29) | 89.49 (+13.49) |
| | Pringles | 64.29 | 73.93 (+9.64) | 73.52 (+9.23) | 66.95 (+2.66) | 81.17 (+16.88) |
| | Tincase | 74.16 | 78.98 (+4.82) | 81.73 (+7.57) | 78.22 (+4.06) | 84.71 (+10.55) |
| | Sandwich | 78.10 | 78.74 (+0.64) | 80.15 (+2.05) | 78.44 (+0.34) | 80.87 (+2.77) |
| | Mouse | 54.54 | 62.59 (+8.05) | 67.12 (+12.58) | 60.98 (+6.44) | 72.49 (+17.95) |
| | Overall | 69.42 | 76.03 (+6.61) | 76.08 (+6.66) | 73.38 (+3.96) | 81.75 (+12.33) |
| B50 | Spray | 76.63 | 84.29 (+7.66) | 82.10 (+5.47) | 83.59 (+6.96) | 89.56 (+12.93) |
| | Pringles | 67.45 | 74.74 (+7.29) | 76.86 (+9.41) | 71.26 (+3.81) | 81.43 (+13.98) |
| | Tincase | 78.33 | 81.24 (+2.91) | 82.13 (+3.80) | 78.42 (+0.09) | 85.00 (+6.67) |
| | Sandwich | 78.38 | 79.18 (+0.80) | 79.90 (+1.52) | 78.59 (+0.21) | 81.54 (+3.16) |
| | Mouse | 54.98 | 62.76 (+7.78) | 67.39 (+12.41) | 60.07 (+5.09) | 72.10 (+17.12) |
| | Overall | 71.16 | 76.44 (+5.28) | 77.68 (+6.52) | 74.39 (+3.23) | 81.93 (+10.77) |
| B75 | Spray | 80.83 | 85.88 (+5.05) | 83.95 (+3.12) | 84.37 (+3.54) | 87.20 (+6.37) |
| | Pringles | 67.48 | 75.38 (+7.90) | 76.83 (+9.35) | 71.02 (+3.54) | 80.31 (+12.83) |
| | Tincase | 75.67 | 80.73 (+5.06) | 82.93 (+7.26) | 77.87 (+2.20) | 84.44 (+8.77) |
| | Sandwich | 78.38 | 78.77 (+0.39) | 80.08 (+1.70) | 78.97 (+0.59) | 80.82 (+2.44) |
| | Mouse | 53.93 | 59.54 (+5.61) | 67.66 (+13.73) | 59.98 (+6.05) | 71.51 (+17.58) |
| | Overall | 71.26 | 76.06 (+4.80) | 78.29 (+7.03) | 74.44 (+3.18) | 80.86 (+9.60) |
| B100 | Spray | 83.59 | 85.20 (+1.61) | 85.59 (+2.00) | 84.02 (+0.43) | 87.22 (+3.63) |
| | Pringles | 69.21 | 74.31 (+5.10) | 76.05 (+6.84) | 71.07 (+1.86) | 80.55 (+11.34) |
| | Tincase | 77.36 | 79.33 (+1.97) | 79.16 (+1.80) | 77.49 (+0.13) | 81.20 (+3.84) |
| | Sandwich | 78.74 | 79.28 (+0.54) | 80.21 (+1.47) | 79.05 (+0.31) | 80.97 (+2.23) |
| | Mouse | 55.80 | 61.61 (+5.81) | 68.22 (+12.42) | 60.49 (+4.69) | 72.66 (+16.86) |
| | Overall | 72.94 | 75.95 (+3.01) | 77.84 (+4.90) | 74.42 (+1.48) | 80.52 (+7.58) |

Table 14: **AUC@[0:0.1] performance under diverse environments of various sensor control methods for FoundationPose.**

| Environment | Object | Baseline RGB: Auto Depth: Default | Depth-Only RGB: Auto Depth: Oracle | RGB-Only RGB: Oracle Depth: Default | Oracle-Fixed Best Fixed Param. | Oracle-Dynamic RGB: Oracle Depth: Oracle |
|---|---|---|---|---|---|---|
| B5 | Spray | 82.80 | 84.68 (+1.88) | 85.34 (+2.54) | 83.44 (+0.64) | 87.17 (+4.37) |
| | Pringles | 25.07 | 39.45 (+14.38) | 45.26 (+20.19) | 32.98 (+7.91) | 64.10 (+39.03) |
| | Tincase | 30.87 | 49.44 (+18.57) | 51.80 (+20.93) | 41.60 (+10.73) | 65.73 (+34.86) |
| | Sandwich | 81.62 | 84.41 (+2.79) | 83.38 (+1.76) | 82.77 (+1.15) | 86.28 (+4.66) |
| | Mouse | 67.88 | 72.80 (+4.92) | 75.10 (+7.22) | 72.02 (+4.14) | 79.54 (+11.66) |
| | Overall | 57.65 | 66.16 (+8.51) | 68.18 (+10.53) | 62.56 (+4.91) | 76.56 (+18.91) |
| B25 | Spray | 84.85 | 86.37 (+1.52) | 87.17 (+2.32) | 84.90 (+0.05) | 88.46 (+3.61) |
| | Pringles | 40.24 | 51.57 (+11.33) | 51.36 (+11.12) | 44.67 (+4.43) | 71.07 (+30.83) |
| | Tincase | 41.24 | 63.33 (+22.09) | 74.76 (+33.52) | 47.56 (+6.32) | 84.11 (+42.87) |
| | Sandwich | 83.36 | 86.72 (+3.36) | 84.90 (+1.54) | 84.10 (+0.74) | 87.97 (+4.61) |
| | Mouse | 71.39 | 75.37 (+3.98) | 75.98 (+4.59) | 72.66 (+1.27) | 80.51 (+9.12) |
| | Overall | 64.22 | 72.67 (+8.45) | 74.83 (+10.61) | 66.78 (+2.56) | 82.43 (+18.21) |
| B50 | Spray | 85.49 | 86.85 (+1.36) | 88.22 (+2.73) | 85.32 (-0.17) | 89.59 (+4.10) |
| | Pringles | 35.67 | 55.31 (+19.64) | 50.74 (+15.07) | 45.50 (+9.83) | 79.00 (+43.33) |
| | Tincase | 47.24 | 63.93 (+16.69) | 71.60 (+24.36) | 54.20 (+6.96) | 82.71 (+35.47) |
| | Sandwich | 83.62 | 87.13 (+3.51) | 85.46 (+1.84) | 85.00 (+1.38) | 88.56 (+4.94) |
| | Mouse | 70.71 | 74.39 (+3.68) | 75.78 (+5.07) | 72.51 (+1.80) | 79.88 (+9.17) |
| | Overall | 64.54 | 73.52 (+8.98) | 74.36 (+9.82) | 68.51 (+3.97) | 83.95 (+19.41) |
| B75 | Spray | 85.93 | 87.20 (+1.27) | 88.73 (+2.80) | 86.12 (+0.19) | 89.83 (+3.90) |
| | Pringles | 35.19 | 49.95 (+14.76) | 60.00 (+24.81) | 54.76 (+19.57) | 81.88 (+46.69) |
| | Tincase | 39.51 | 59.67 (+20.16) | 78.89 (+39.38) | 52.82 (+13.31) | 86.93 (+47.42) |
| | Sandwich | 83.56 | 86.82 (+3.26) | 85.82 (+2.26) | 84.92 (+1.36) | 88.92 (+5.36) |
| | Mouse | 70.98 | 74.78 (+3.80) | 75.59 (+4.61) | 72.88 (+1.90) | 80.10 (+9.12) |
| | Overall | 63.03 | 71.68 (+8.65) | 77.81 (+14.78) | 70.30 (+7.27) | 85.53 (+22.50) |
| B100 | Spray | 84.07 | 87.51 (+3.44) | 88.54 (+4.47) | 86.00 (+1.93) | 90.00 (+5.93) |
| | Pringles | 35.14 | 57.31 (+22.17) | 46.55 (+11.41) | 48.07 (+12.93) | 77.40 (+42.26) |
| | Tincase | 38.89 | 55.69 (+16.80) | 73.11 (+34.22) | 57.33 (+18.44) | 85.27 (+46.38) |
| | Sandwich | 83.69 | 86.92 (+3.23) | 86.03 (+2.34) | 85.21 (+1.52) | 88.82 (+5.13) |
| | Mouse | 70.17 | 74.37 (+4.20) | 76.88 (+6.71) | 72.46 (+2.29) | 79.85 (+9.68) |
| | Overall | 62.39 | 72.36 (+9.97) | 74.22 (+11.83) | 69.81 (+7.42) | 84.27 (+21.88) |

# H    THE USE OF LARGE LANGUAGE MODELS (LLMs)

In this work, editing, data processing, and visualization were polished with the assistance of ChatGPT (OpenAI, 2025) and Gemini (Google, 2025).

