# OpenReview forum: "SenseShift6D: Multimodal RGB-D Benchmarking for Robust 6D Pose Estimation across Environment and Sensor Variations"
_ICLR.cc/2026/Conference — ICLR 2026 Conference Desk Rejected Submission_

### Official Review · Reviewer_bar8 · 2025-10-29

**Soundness:** 2
**Presentation:** 2
**Contribution:** 3
**Rating:** 4
**Confidence:** 3

**Summary:**

The paper introduces SenseShift6D, a new real-world RGB-D benchmarking dataset for 6D pose estimation that emphasizes variation in sensor and environmental settings (exposure, gain, illumination, and depth). The dataset features 5 objects, 166.4k RGB frames, 16.7k depth frames, and a large number of sensor/lighting permutations (reported as 1,380 unique sensor–lighting permutations per object pose) and is evaluated with both pretrained, generalizable pose estimators and instance-level models. The experiments analyze the effect of autoexposure and sensor configurations and show that multimodal (RGB+depth) test-time sensor adaptation can improve performance; the authors demonstrate that an optimal test-time sensor adaptation yields an AUC improvement of at least +15 over baseline auto-exposure settings.

**Strengths:**

-	Introduction of an extensive 6D pose dataset that incorporates diverse illuminations, exposure, gain and depth levels. This enables benchmarking the robustness pose estimation models in challenging real-world conditions.
-	Comprehensive evaluation on different sensor configurations with pretrained models as well as instance-level pose estimation models.
-	The AUC with optimal sensor control is at least +15 points higher compared to the baseline auto-exposure sensor configurations, indicating the potential of test-time sensor control (Table 4).

**Weaknesses:**

-	There is no validation split, instead there is only a training and test split, which might lead to potential overfitting of the evaluated models.
-	The motivation is sensor-aware test-time adaptation, but the paper only shows the Oracle upper bound and not a single practical adaptation method.
-	Limited number of objects in the dataset (5 objects) may lead to non-generalizable conclusions.

**Questions:**

-	How do you concretely obtain the Oracle selection in the evaluation? Wouldn’t there be a validation split necessary for such a procedure?
-	In Table 5, training on Train-Var does not result in a performance gain on Test-Var for some object categories. Do you have any insights on that?

---

> ### Author Response · Authors · 2025-11-21
>
> **W1 + Q1 – Lack of validation split**
> - Regarding the absence of an official validation split, our design follows **common practice in 6D pose benchmarks** (e.g., LM/LM-O, T-LESS, YCB-V), which typically provide train + test splits and leave construction of a validation subset to the user or to challenge organizers. In the BOP ecosystem, separate “validation” splits are usually introduced by challenge protocols primarily because test labels are hidden, not because the datasets themselves require a fixed validation partition. Therefore, we chose to **devote our expensive real captures** to maximizing coverage of training and test across sensor/illumination conditions, rather than reserving a fixed validation subset. Users who require a validation split (e.g., for new methods) can **easily derive** one from Train-Def/Train-Var (e.g., by holding out specific poses, scenes, or sensor settings), as is standard practice with other pose datasets.
> - Both Oracle methods are not trained and do not require validation split.
> (1) **Oracle-Dynamic:** for each test scene we evaluate the model on all available sensor configurations and mark the scene correct if any configuration yields a correct pose under the ADD threshold, using ground truth. It is intended strictly as an upper bound on what an ideal per-scene sensor policy could achieve.
> (2) **Oracle-Fixed:** we compute performance for each RGB-D configuration across all test scenes and report the best-performing single configuration as a global upper bound for any fixed setting. It is used only for analysis (to show the gap between fixed vs. adaptive settings), not for training or hyperparameter tuning.
>
> **W2 – No practical sensor-adaptation method**
> - We agree that our work does not propose a practical online sensor-control algorithm. Our scope is to **quantify and expose** the potential of sensor-aware test-time adaptation and to provide a benchmark that makes such study possible. Oracle-Dynamic answers the question of how much performance could be gained if an ideal policy always chose the best sensor configuration for each scene, without retraining. The large gains we observe (+19.5 pp AUC for pretrained models and consistent gains for instance-level models) indicate that sensor parameters are a **powerful but underused degree of freedom.**
> - Designing an actual control policy is a **separate research problem**. A natural idea is to use model feedback (e.g., confidence scores, reprojection errors, RGB–depth consistency, or multi-view checks) to guide sensor selection, but current pose estimators generally lack calibrated, comparable confidence estimates across frames and sensor settings. Building such a confidence module and integrating it with the sensor pipeline involves trade-offs between robustness, computation, and latency. We will briefly discuss these directions and position SenseShift6D explicitly as a **foundation and testbed for future practical sensor-control algorithms**, rather than a solution to online control itself.
>
> **W3 – Limited number of objects**
> - SenseShift6D contains five objects and single-object tabletop scenes by design. Rather than combining all difficulties (many objects, clutter, occlusion), we chose to **go very deep along a new axis**—physically captured sweeps over illumination, exposure, gain, and depth mode. Each pose is captured under up to **1,380 sensor–lighting permutations**, for a total of 166.4k RGB, 16.7k depth, and **665.9k valid RGB-D scenes** (the **largest** to date, to our knowledge). Even with only five objects, Baseline performance **varies markedly** between them, and sensor control substantially reduces this gap, showing that illumination and sensor variation alone already form a challenging and practically relevant regime. We view SenseShift6D as **complementary** to object- and clutter-focused datasets such as LM/LM-O, T-LESS, YCB-V, HOPE, and IPD.
> - Meanwhile, we have already **extended** SenseShift6D by collecting an additional object (a rubber duck), increasing the totals to 198.8k RGB, 20.0k depth, and **795.2k RGB-D scenes**. We will report experimental results on this extension soon.
>
> **Q2 - Train-Var not always improving performance**
> - As the reviewer pointed out, regarding Table 5, training on **PBR + Train-Var** does not improve performance on Test-Var for some objects. For example, the mouse object has an ambiguous shape and an almost textureless surface, providing only weak RGB cues even under good sensor settings. When trained with data that contains large appearance variation—especially scenes captured with severely suboptimal sensor parameters (e.g., very low exposure, extreme gain)—these cues are **degraded or lost at the measurement stage** by illumination changes, gain fluctuations, and sensor noise. As a result, the model is asked to **"learn from missing information,"** struggles to learn stable, object-specific features and may instead latch onto unstable signals tied to sensor conditions.

---

### Official Review · Reviewer_rQYE · 2025-11-01

**Soundness:** 3
**Presentation:** 3
**Contribution:** 2
**Rating:** 4
**Confidence:** 4

**Summary:**

This paper presents SenseShift6D, a new benchmark including RGB and Depth images for 6D pose estimation. The benchmark encompasses a range of image capture setups, including RGB exposures, RGB gains, as well as depth capture modes and illumination levels for five household objects. The benchmark expands the existing 6D pose estimation dataset by incorporating additional environmental and sensor-related variations. The evaluation of a popular estimation algorithm on the benchmark shows the potential of the SenseShift6D and identifies the new challenges in current 6D pose estimation.

**Strengths:**

1.	The paper is well written and structured.
2.	The paper is well-motivated and contributes meaningfully to the 6D pose estimation research.
3.	The extensive evaluation of existing algorithms on the benchmark presents many valuable insights.

**Weaknesses:**

1.	The collected dataset has the ChArUco board presented in the images, also for the other well-known dataset. The question is whether the ChArUco board could potentially introduce bias to the pose estimation algorithms if the estimation algorithm is trained on a dataset that includes the ChArUco board. See more detailed discussion in [1]. I suggest removing the calibrator after calibration is done, as in the YCB dataset.
2.	The number of objects and the background of the objects are limited. The multi-object with a more complex background and a more cluttered environment is missing, as discussed in the limitations.
3.	Although the evaluation on the test dataset with sensor configuration adaptation shows the potential of improving the detection performance,  the current manual tuning procedure is not applicable in the real application, as the correct prediction is unknown. From this perspective, it is not entirely clear that adaptive sensor control on demand is a viable practical approach to addressing the challenges of environmental factor shifts.

[1] Uncovering the Background-Induced bias in RGB based 6-DoF Object Pose Estimation, https://arxiv.org/abs/2304.08230

**Questions:**

Could you please discuss the potential methods of online automated sensor control and the potential problems (limitations) associated with it?

---

> ### Author Response · Authors · 2025-11-21
>
> **W1 – ChArUco and potential bias**
>
> We appreciate the pointer to [1] and agree that static calibration patterns can introduce bias. As in established pose datasets, SenseShift6D uses a ChArUco board as a calibration target to obtain accurate, repeatable 6D ground truth while we sweep many sensor settings.
> - Pretrained generalizable models (GigaPose, SAM-6D, FoundationPose) are never trained on SenseShift6D and are evaluated with ground-truth masks, so **the board cannot become a learned shortcut for them.**
> - Instance-level models (ZebraPose, GDRNPP, HiPose) are trained on SenseShift6D and could in principle exploit the board. To probe this, we additionally evaluated them on a modified Test-Var split where the board region is painted black. Only the RGB-only methods (ZebraPose, GDRNPP) show noticeable, object-dependent changes between the original and board-removed images, whereas the RGB-D model (HiPose) remains essentially unchanged, suggesting that **multimodal inputs mitigate reliance on RGB-only background cues**. We will report these results in Appendix E.
>
> Digitally erasing the board also introduces artificial artifacts that can suppress genuine sensor noise and illumination effects, so we do not plan to distribute a marker-removed version of the dataset. Instead, we will provide accurate masks for the board region, allowing users to remove, inpaint, or replace the markers according to their own needs and processing pipelines.
>
> **W2 – Few objects and simple background**
>
> - We agree that our dataset has fewer objects and simpler scenes than HOPE or YCB-V. This is **deliberate: our aim is not to combine all challenges, but to study a largely unexplored axis**—physically realized variation in illumination and multimodal sensor parameters (exposure, gain, depth mode)—in a clean, controlled way. Existing pose benchmarks either fix camera settings or vary only a few exposure levels; none jointly sweep exposure, gain, and multiple depth modes, so they cannot isolate robustness to sensor-domain shifts.
> - We therefore **go very deep along this axis**: each pose is captured under a dense grid of sensor–lighting permutations, yielding 166.4k RGB, 16.7k depth, and **665.9k valid RGB-D scenes (1,380 RGB-D scenes per object pose)**. Even with only 5 objects, Baseline performance **varies widely** across objects, and Oracle-Dynamic multimodal control significantly improves AUC and reduces object- and scene-wise variance. Instance-level models trained on the same objects and background also drop from Test-Def to Test-Var, and oracle test-time sensor control often outperforms strong RGB augmentation and training on a much larger physically varied set.
> - These results show that **illumination and sensor variation alone are already a substantial challenge**. We will clarify that SenseShift6D is a focused, complementary benchmark to cluttered, multi-object datasets, which primarily target object/scene diversity under fixed sensor settings.
> - Meanwhile, we have expanded SenseShift6D by collecting another object (a rubber duck), now resulting in 198.8k RGB, 20,0k depth, and 795.2k RGB-D scenes. We will report experimental results on the additional data soon.
>
> **W3 + Q – Scope, practicality and potential methods**
>
> - Oracle-Dynamic simply checks, using ground truth, for each scene (pose/object/illumination) whether there **exists at least one RGB-D configuration** that yields a correct prediction for a given model; it does not address how to find that configuration in practice. As a dataset-and-benchmark paper, our goal is to provide a benchmark that **(1) quantifies this potential** (an upper bound) and **(2) offers the dense multimodal sweeps** needed to develop and evaluate future sensor-control mechanisms.
>
> - The development of online sensor control requires **additional novel work**. A natural idea is to use feedback from the pose estimator—for example, maximizing a confidence score. However, most existing 6D pose estimators do not expose a reliable, calibrated confidence that can be compared across sensor settings or frames; current generalizable models rely on coarse-stage scores, and FoundationPose uses internal ranking scores within one frame, none of which are designed as absolute confidences for closed-loop control. A promising direction is therefore to develop an auxiliary, model-agnostic confidence module (e.g., based on reprojection error, RGB–depth consistency, or learned calibration) that can be applied across different pose estimators and sensor settings. Introducing such a module and control loop adds computational overhead and latency, so real-time systems will require careful co-design of the sensor and inference pipelines.
>
> - We will clarify this scope and expand the discussion on the development of online sensor control, positioning our work explicitly as a **foundation and testbed for future practical sensor-control algorithms**, rather than a solution to online control itself.

---

### Official Review · Reviewer_KbWb · 2025-11-02

**Soundness:** 4
**Presentation:** 4
**Contribution:** 4
**Rating:** 6
**Confidence:** 3

**Summary:**

This paper presents a new benchmark dataset for 6D object pose estimation named SenseShift6D. It contains data with various sensor parameters including RGB exposure, gain and depth capture modes, as well as various environmental conditions such as illumination levels. The dataset contains 166.4k RGB and 16.7k depth images of 5 household objects. The authors found that test-time sensor control by dynamically selecting optimal sensor parameters at inference can improves object pose estimation accuracy.

**Strengths:**

- The idea of incorporating several camera parameter variations in object pose dataset collection is nice.
- This benchmark explores RGBD sensor parameter variation (especially photometric parameters) for 6D pose estimation. This is an interesting and important aspect of the object pose estimation problem.
- The benchmark captures real camera effect under different parameter variation. This effect is not easily re-produced in synthetic data. This will be valuable for studying the problem.
- The dataset has a clear and well-defined split: Train-Def, Train-Var, Test-Def, and Test-Var.
- It provides benchmarking of several recent methods.

**Weaknesses:**

- The dataset is not marker-less. There are a lot of AR tags on the board beneath the object. It is OK since other datasets such as LineMOD also did this. But this dataset would still be far away from "data in the wild".
- This dataset only contains 5 objects which is quite small in number compared to other related datasets.
- It only contains single-object tabletop scenes without occlusion or multi-objects scenes.
- Not sure if all RGB and depth sensors can control their modes to allow varying camera parameters. For RGB and depth cameras who parameters are not easily changeable at test time, the contribution of this dataset would be significantly reduced.

**Questions:**

- The RGB and depth images does not have the same number of images. Do RGB images and depth images have one-on-one correspondences?
- Is the Sandwich object rigid or deformable? If deformable, how to ensure its shape stays the same?
- How to convince users to use this dataset instead of other related datasets such as HOPE and IPD? They contain more objects anyway.

---

> ### Author Response · Authors · 2025-11-21
>
> **W2, W3, Q3 – Few objects, simple scenes, and relation to other datasets**
>
> - Our aim is not to build the “hardest” dataset by combining all challenges (object diversity, clutter, occlusion), but to open up **a new axis of difficulty**: physically realized illumination and sensor sweeps for RGB and depth. To study this axis cleanly, we deliberately keep the rest of the scene simple: five objects, single-object tabletop, and a fixed background. Although there are only five objects, each pose is captured under up to **1,380 physical sensor–lighting permutations**, yielding **665.9k valid RGB-D scenes** (the **largest** count we are aware of). SenseShift6D therefore goes **very deep** along the sensor/environment axis instead of going wide over many objects and backgrounds. Adding heavy clutter, occlusion, and many more objects now would require substantial effort before we even understand how models behave along this axis or which sensor-control strategies are effective.
>
> - Even in this controlled regime, current methods already struggle: for pretrained generalizable models, Baseline AUC varies by ~40 pp between objects, and per-scene multimodal sensor control improves overall AUC by +19.5 pp while reducing object- and scene-level variance. Instance-level models trained on the same objects and background also degrade from Test-Def to Test-Var, and oracle test-time sensor control often outperforms both heavy RGB augmentation and training on a 3.5× larger physically varied set. Illumination and sensor variation alone already constitute a significant and practically relevant challenge.
>
> - We therefore see SenseShift6D as **complementary** to existing datasets: the LineMOD family, T-LESS, and YCB-V for cluttered or occluded scenes, and HOPE/IPD for household and industrial objects, all captured under **fixed or weakly varied** sensor settings. For HOPE and IPD you mentioned, HOPE varies illumination but **not** sensor parameters; and IPD varies exposure at **only four** levels and does **not** touch RGB gain or depth modes (number of per-pose variations is **12 in IPD vs. 1380 in ours**). While they remain ideal for studying object and scene diversity, SenseShift6D is intended for studying robustness to sensor/lighting shifts and test-time sensor control. We will emphasize this complementary role and note that extending the benchmark to cluttered, multi-object settings is a natural next step.
>
> - Meanwhile, to begin addressing the object-diversity concern, we have **already expanded** SenseShift6D by collecting an **additional object (a rubber duck)**, bringing the dataset to 198.8k RGB, 20.0k depth, and **795.2k RGB-D scenes**. We will report experimental results on this extended data soon.
>
> **W1 – Markers / not “in the wild”**
>
> - We agree and do not claim that SenseShift6D itself is “in the wild.” Instead, our goal is to take a **necessary step** towards models that operate robustly in in-the-wild environments by capturing an **underexplored challenge**—-variation in sensor and environmental conditions. To study this axis, we follow common practice in pose datasets (e.g., LM/LM-O, T-LESS) and use a rigid marker board in a controlled setup to obtain accurate, repeatable 6D ground truth while sweeping a dense grid of settings. We also provide pixel-accurate masks so users can crop or mask out the board if they wish to avoid any marker cues.
>
> **W4 – Not all cameras expose sensor controls**
>
> - Importantly, our work is **fundamental research**: it empirically demonstrates the impact of lighting and sensor variations and the value of test‑time sensor control for robust inference. Practical implementation barriers today should not diminish that value; **once the benefits are clear**, hardware and APIs often **evolve** to expose more control to practitioners. We will emphasize this perspective. In addition, for the less-controllable devices the reviewer mentioned, our dataset can still be useful as a **diagnostic tool** to quantify model sensitivity to environmental changes and to guide sensor-robust training.
>
> **Q1 – RGB vs. depth counts**
>
> - RGB and depth are **consistently pairable** even though the raw image counts differ. For each pose and illumination level, we keep the camera and object **fixed** and sweep through all RGB settings and all depth presets in the split, yielding a valid RGB-D scene with identical 6D ground truth. The number of depth scenes is less than that of RGB scenes simply because there are a limited number of depth presets.We encode this by sharing the same scene IDs and image indices across RGB and depth, which allows each RGB image to be paired with the corresponding depth capture.
>
> **Q2 – deformable?**
>
> - The sandwich is made of a sponge-like material, always placed gently on the ChArUco board, and **not manipulated or pressed** during acquisition. We verified visually that its shape remains stable across poses, and we treat it as **approximately rigid** for annotation and evaluation.

---

### Author Response · Authors · 2025-11-21

We thank Reviewer KbWb (R1), Reviewer rQYE (R2), Reviewer bar8 (R3) for the thoughtful and constructive feedback. We deeply appreciate that:
- R1, R2, R3 all found the paper **well‑written and well‑structured**, and viewed the benchmark as a meaningful contribution to 6D pose estimation.
- R1, R2 highlighted the **clear split** design (Train‑Def/Var, Test‑Def/Var) and the **focus** on sensor/illumination variation.
- R1, R2, R3 valued the **extensive** experiments on both pretrained generalizable models and instance‑level models, and R3 specifically emphasized the **large performance gains** from multimodal sensor control.

These strengths reflect our main goal: to expose a new, sensor‑aware axis of robustness and quantify the impact of test‑time sensor control.

Main concerns and our brief clarifications are as below. The detailed response for each reviewer is put as comments.

**Few objects, simple scenes, relation to HOPE/IPD/YCB‑V (R1, R2, R3)**
- We clarified that SenseShift6D is intentionally focused: it goes deep along a new axis—physically captured variation in illumination, exposure, gain, and depth mode (166.4k RGB, 16.7k depth, 665.9k RGB‑D scenes)—while keeping scenes simple to isolate sensor effects. Existing datasets emphasize object/scene diversity under fixed sensors; SenseShift6D is designed as a complementary benchmark for sensor/lighting robustness and test‑time control. During rebuttal we also extended the dataset with a sixth object (rubber duck), bringing it to 198.8k RGB, 20.0k depth, and 795.2k RGB‑D scenes.

**ChArUco board and background‑induced bias (R1, R2)**
- We use the board, as in many pose datasets, to obtain accurate 6D ground truth while sweeping a dense sensor grid. Pretrained models are not trained on SenseShift6D and are evaluated with GT masks, so they cannot exploit the board. For instance‑level models we added a new experiment on Test‑Var with the board region painted black: RGB‑only models show some object‑dependent changes, whereas HiPose (RGB‑D) is essentially unchanged, suggesting multimodal inputs mitigate reliance on RGB backgrounds. Rather than releasing a globally “board‑erased” version (which would distort sensor noise), we will provide board masks so users can crop/inpaint as they wish.

**Oracle sensor control vs. practical policies; sensor controllability (R1, R2, R3)**
- R2 and R3 correctly note that Oracle‑Dynamic is not deployable. Its purpose is to define an upper bound: for each scene, does there exist a sensor configuration that makes it easy for the given model? The large gains (+19.5 pp AUC for pretrained models, consistent gains for instance‑level models) show that sensor parameters are a powerful but underused degree of freedom. Designing a practical online policy (e.g., confidence‑based or RGB‑D/multi‑view‑consistency‑based) is a separate research problem, complicated by the lack of calibrated confidence scores and by compute/latency constraints. We now emphasize that SenseShift6D is a foundation and testbed for such methods, not a complete online control solution. Regarding R1’s concern that some cameras expose limited controls, we position this as fundamental research: demonstrating the value of sensor control via RGB‑D sensors that already support the parameters we vary can motivate broader API support over time.

**Validation split and potential overfitting (R3)**
- Our split (Train‑Def/Var vs. Test‑Def/Var) follows common practice in 6D pose benchmarks such as LM/LM‑O, T‑LESS, and YCB‑V, which typically provide train+test and leave validation to users or challenge organizers. Pretrained models are not tuned on our data; instance‑level models reuse their original hyperparameters (no search on our test sets), and train/test sensor configurations are disjoint. Given the cost of dense real capture, we prioritized coverage across sensor conditions; users can easily derive a validation subset from Train‑Def/Var if needed.

**Train‑Var not always improving Test‑Var (R3)**
- As we explain, Train‑Var deliberately includes many frames captured under severely suboptimal sensor settings, where key cues are lost at measurement time; no amount of training can recover missing information. Test‑time sensor control instead avoids such bad measurements and steers the camera toward information‑rich configurations, explaining why Oracle control can outperform Auto even with Train‑Var and why training and sensor control are complementary.

We again thank R1, R2, and R3 for their insightful comments and for recognizing the strengths of our work. We believe the clarifications, new experiments (board‑masked evaluation), and dataset extension further strengthen the paper and reinforce SenseShift6D as a focused, sensor‑aware complement to existing 6D pose benchmarks. We will incorporate these aspects in the revised manuscript and upload it.

---

### Note · Program_Chairs · 2026-01-17
**Submission Desk Rejected by Program Chairs**

The following references in this submission do not refer to real documents and/or have major errors in bibliographic information:

 Martin Sundermeyer, Tomas Hodan, Bertram Drost, Suat Gedikli, David Winkelbauer, Zoltan-Csaba Marton, Matthias Brucker, Nicolas Zeller, and Rudolph Triebel. Hope: A household objects dataset for pose estimation. In Proceedings of the IEEE/RSJ International Conference on Intelligent Robots and Systems (IROS), pp. 9664-9671, 2022